# Energy and climate policy implications on the deployment of low-carbon ammonia technologies

Chi Kong Chyong[1,2] ✉, Eduardo Italiani[2] & Nikolaos Kazantzis[3]

The economic feasibility of low-carbon ammonia production pathways, such as steam methane reforming with carbon capture and storage, biomass gasification, and electrolysis, is assessed under various policy frameworks, including subsidies, carbon pricing, and renewable hydrogen regulations. Here, we show that employing a stochastic techno-economic analysis at the plant level and a net present value approach under the US Inflation Reduction Act reveals that carbon capture and biomass pathways demonstrate strong economic potential due to cost-effectiveness and minimal public support needs. Conversely, the electrolytic pathway faces significant economic challenges due to higher costs and lower efficiency. We conclude that efficient decarbonization of ammonia production requires adapting the Haber-Bosch process for variable bioenergy quality, ensuring safe $CO_2$ transport and storage, advancing research to lower costs and improve efficiency in renewable energy and storage technologies, as well as creating a technologically neutral policy framework.

Ammonia is a pivotal energy vector in the ongoing global energy transition, serving as a versatile feedstock and a prospective low-carbon fuel for diverse applications, including electricity and maritime transport[1,2]. Its role is further amplified by its ability to be a reliable storage and transport medium for low-carbon hydrogen. Further, ammonia benefits from an established global market and relatively mature infrastructure. Unlike hydrogen, the pipeline and shipping technology for transportation of ammonia is mature and has a market size upwards of $70 (U.S. Dollars, $) billion[3]. Yet, ammonia accounts for a substantial 3% of global $CO_2$ emissions, with a carbon intensity that outpaces even steel and cement[1,4]. The prevailing ammonia production (AP) pathway, reliant on steam methane reforming (SMR) and the Haber-Bosch (HB) process, is predominantly fossil-fuel-based[5]. A staggering 98% of current production is sourced from fossil fuels, namely natural gas and coal. The technological interplay between hydrogen and ammonia production offers promising avenues to decarbonize traditionally hard-to-abate sectors. Ammonia's storage and transport economics are advantageous, requiring milder conditions (-33 degrees Celsius (C) or 10–20 bar for ammonia versus -253C or

350–700 bar for hydrogen) than hydrogen[3]. This multi-dimensional utility positions ammonia as a critical energy vector in a resilient, low-carbon energy system. However, its decarbonization remains a formidable challenge, necessitating technological and policy interventions to mitigate its environmental impact.

The high costs of new, low-carbon technologies, their nascent development and limited demonstration at a commercial scale, and the lack of policy support have hindered investments and widespread use. In 2021, renewable ammonia production was just 0.02Mt (megatonnes)[2]. By 2025, new AP projects are expected to primarily utilize low-carbon methods[2].

The Inflation Reduction Act (IRA) offers subsidies to scale up low-carbon energy technologies in the United States. The IRA offers substantial tax credits, grants, loans, and rebates for carbon capture, low-carbon hydrogen production, and energy generation[6]. However, the IRA also introduces inherent risks for deploying low-carbon ammonia technologies, mainly due to increased dependence on grid electricity emissions and prices, as these emerging technologies tend to be more electrically intensive than conventional SMR[7].

[1]Oxford Institute for Energy Studies, Oxford, UK. [2]Center on Global Energy Policy, School of International and Public Affairs, Columbia University, New York, NY, USA. [3]Department of Chemical Engineering, Worcester Polytechnic Institute, Worcester, MA, USA. ✉e-mail: Kong.Chyong@oxfordenergy.org

The International Energy Agency (IEA) notes that 60% of ammonia emission reductions are expected from technologies in the demonstration phase[1]. The success of these low-carbon technologies may depend on the grid's energy transition and advancements, as well as cost reductions in green technologies like wind, solar, batteries, and electrolysis during and after the IRA's time frame. This reliance, along with the long lifespans of AP plants (around 40 years), puts potential low-carbon ammonia production investments at significant risk of loss, especially when competing against AP SMR.

AP SMR capitalizes on the existing robust natural gas infrastructure and highly mature technology, providing it with a significant economic edge over emerging and risky technologies[1]. The inclusion of AP SMR as a benchmark in our analysis, an important detail not covered in every study, establishes a high benchmark for alternative technologies aiming to enter the ammonia commodity market.

This study evaluates the economic impacts of the IRA on low carbon ammonia production technologies, focusing on technology, policy, and market uncertainties. We employ a stochastic Discounted Cash Flow model to assess the IRA's financial provisions and risks for low carbon ammonia production: conventional SMR, SMR with carbon capture systems (CCS), indirect biomass gasification coupled with SMR (BH2S), and alkaline electrolysis (AEC). Unlike traditional discounted cash flow models that rely on deterministic cash flow estimates, our approach incorporates uncertainties, addressing the "flaw of averages"[8] in the complex, non-linear AP value chain. The discounted cash flow approach will look at three scenarios. Scenario A assumes the grid electricity is used to power the low carbon ammonia production technologies and SMR. Scenarios B and C use renewable wind, solar energy, and battery storage to power the low carbon ammonia production technologies. In particular, Scenario B represents the case where the AP plant owns the generation facility. Scenario C relies on a power purchase agreement (PPA) between the AP and the renewable energy facility.

## Results

### Implications of the IRA on low-carbon ammonia deployment

Our modeling results highlight that the successful deployment of low carbon ammonia production under the IRA depends on lifecycle carbon intensity (CI) of not just feedstock (natural gas and biomass) but, crucially, electricity (Fig. 1). AP SMR needs 64 MW of electricity (MWe)

to operate, whereas AP CCS, AP BH2S, and AP AEC require 117 MWe, 127 MWe, and 910-1010 MWe, respectively. Consequently, low carbon ammonia production technologies, especially AEC, are more sensitive to electrical lifecycle carbon emissions. We find the IRA framework does not reward (enough) low carbon ammonia production technologies connected to the US power grid (Scenario A) - although expected to decarbonize significantly under the IRA, the grid is still carbon-intensive to the extent that the subsidies are not matched with the marginal cost of reducing carbon emissions through low carbon ammonia production, making the incumbent technology - SMR - always economically a better choice for investors (Fig. 2, Scenario A). This conclusion holds under two time periods analyzed - 2026 and 2033 - to account for technology improvements and cost reductions. It holds across thousands of Monte Carlo simulations covering critical variables that determine the economics of these technologies (see Supplementary Methods).

Then, the critical question this paper aims to answer is under what conditions the IRA will likely stimulate the deployment of low carbon ammonia production. We find that only when low carbon ammonia production's electricity consumption is carbon-free will we likely witness their economics outperform the SMR's. We distinguish between a vertically integrated business model where an AP investor would build and own an off-grid hybrid renewable (wind and solar with battery storage) electricity farm (Scenario B) and a case in which the investor signs a long-term PPA with the hybrid renewable farm—allowing for a fixed electricity price at the farm's levelised cost of electricity (LCOE) (Scenario C). In the Scenario B, investors anticipate combining either 48E or 45Y credits, and the sale of surplus electricity from the hybrid renewable farm will result in a net positive cash flow. Conversely, AP investors in the PPA (C) scenario opt for a long-term PPA at a fixed price, equating to the LCOE required to supply firm renewable power to their AP plant, thereby avoiding the upfront capital investment risk of owning the hybrid farm. Scenarios B and C allow low carbon ammonia production technologies to qualify for the highest 45V tranche ($3 kg-1 under 0.45 KgCO2eq KgH2-1) because electrical emissions are negligible compared to the CI of electricity from the grid (Scenario A).

Our results show that under the PPA, the Net Present Value (NPV) of CCS and BH2S is the highest and outperforms SMR in almost all the simulations and years considered. Albeit consuming

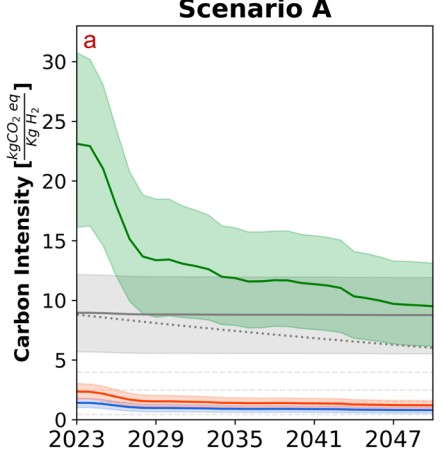
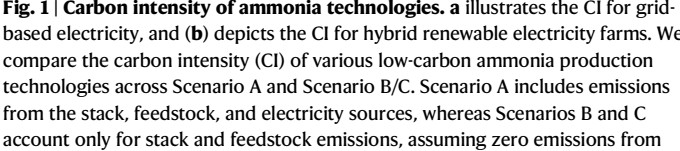
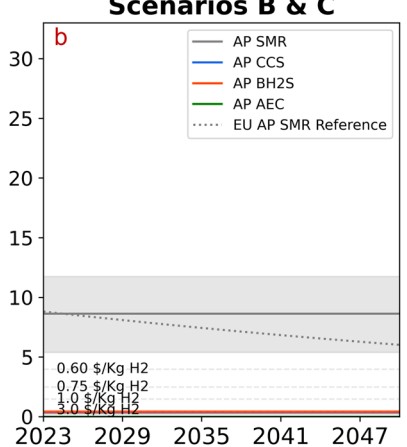

**Fig. 1 | Carbon intensity of ammonia technologies. a** illustrates the CI for grid-based electricity, and (**b**) depicts the CI for hybrid renewable electricity farms. We compare the carbon intensity (CI) of various low-carbon ammonia production technologies across Scenario A and Scenario B/C. Scenario A includes emissions from the stack, feedstock, and electricity sources, whereas Scenarios B and C account only for stack and feedstock emissions, assuming zero emissions from

renewable energy sources. Symbols used: light gray line represents AP SMR, blue line represents AP CCS, orange line represents AP BHS, green line represents AP AEC, and the dotted gray line serves as a reference for the European Union's AP SMR standards. Horizontal threshold lines indicate the 45V tax credit thresholds. Uncertainty bands reflect the range of potential carbon intensities, with the EU AP SMR treated as deterministic.

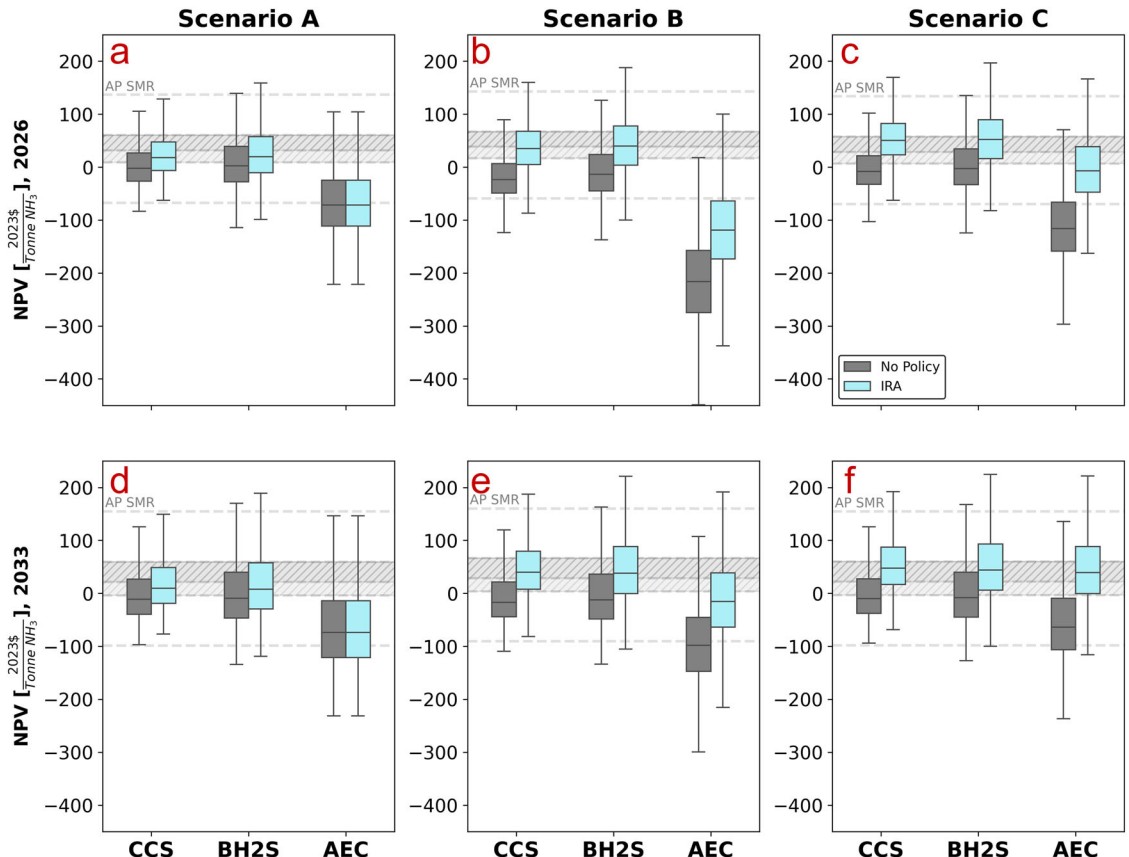

**Fig. 2 | Net present value of ammonia technologies. a–c** depict the NPVs for Scenario A using grid electricity, Scenario B with colocated wind and solar farm, and Scenario C with a power purchase agreement with a wind and solar farm for the year 2026, respectively. (**d–f**) represent the same scenarios projected for the year 2033. We compare the net present value (NPV) of various low-carbon ammonia production technologies against a control scenario with no policy support and ammonia production through steam-methane reforming (AP SMR). Scenario B involves colocated wind and solar farms directly integrated with the ammonia plant, allowing for direct control and optimization of renewable energy generation, whereas Scenario C involves a power purchase agreement (PPA) with external wind and solar farms. Symbols used: light blue data points represent NPVs with the Inflation Reduction Act (IRA), and gray data points represent NPVs without the IRA. AP SMR refers to Ammonia Plant using Steam Methane Reforming, AP CCS to Ammonia Plant with Carbon Capture and Storage, AP BHS to Ammonia Plant using Biomass Gasification coupled with Steam Methane Reforming, and AP AEC to Ammonia Plant using Alkaline Electrolysis. Y-axis ranges vary across scenarios to display the full bar chart for each technology.

electricity with zero carbon emissions, the subsidies are insufficient to justify the upfront capital expenditure (CAPEX) for AP investors to build and own the off-grid hybrid renewable farm in the near term (2026) because the near-term cash flow highly influences the NPV. By 2033, the economics of CCS and BH2S will likely outperform SMR under the scenario B. Nevertheless, scenario C offers the highest NPV for CCS and BH2S. Furthermore, the scenario B business model for AEC is particularly unattractive due to its substantially higher electricity requirement—approximately eight times that of the other technologies like CCS and BH2S (See Supplementary Table 7) - resulting in a CAPEX for power generation that is eightfold larger.

AEC economics heavily depends not just on subsidies but crucially on wind, solar and electrolysis cost reductions. In the near term (2026), AEC will unlikely deliver higher NPV (than SMR) in all configurations considered. Only, in 2033, AEC, with IRA subsidies and under the PPA arrangement, offers a 50% higher return (median NPV) than SMR. Most of AEC's economic improvements between 2026 and 2033 include technological cost reduction and wind and solar resource improvements. However, CCS and BH2S pathways offer higher NPV (relative to SMR) than the AEC pathway in the near (2026) and medium term (2033). CCS and BH2S offer 34–58% higher economic returns than SMR (in 2033, Scenario C), thus outpacing the AEC pathway not just in 2033 but in the near term.

## Cost of supporting low-carbon ammonia deployment

The IRA framework will likely stimulate investment to decarbonize the US AP through generous subsidies. An important policy question is understanding the trade-off between low carbon ammonia production's economics and their carbon abatement costs, CAC (Fig. 3). From a public policy perspective, the low carbon ammonia production technologies that achieve the highest NPV (relative to the incumbent SMR) at the lowest cost to society should be supported. All three low carbon ammonia production technologies achieve higher NPV than the SMR under scenario C (Fig. 2, Scenario C). Under the IRA, relevant subsidy schemes (see Supplementary Methods) are stackable. Therefore, the CAC is fully reflected in Scenario B, where total subsidy costs include low carbon ammonia production and renewable energy generation support.

Thus, the CAC of the CCS and BH2S pathways is substantially lower than that of the AEC pathway. In most simulations, their CAC does not exceed the social cost of carbon and is only marginally higher than the 2020-22 range of the EU carbon price. In 2033, when AEC's economics exceed the baseline SMR, its CAC (median value) is ca. $162 per tCO2e, 56% higher than the CAC of BH2S and 29% higher than CCS' CAC.

The IRA framework incentivizes investments in low carbon ammonia production technologies through tax credits. The policy framework now offers direct payment and transferability of tax credit

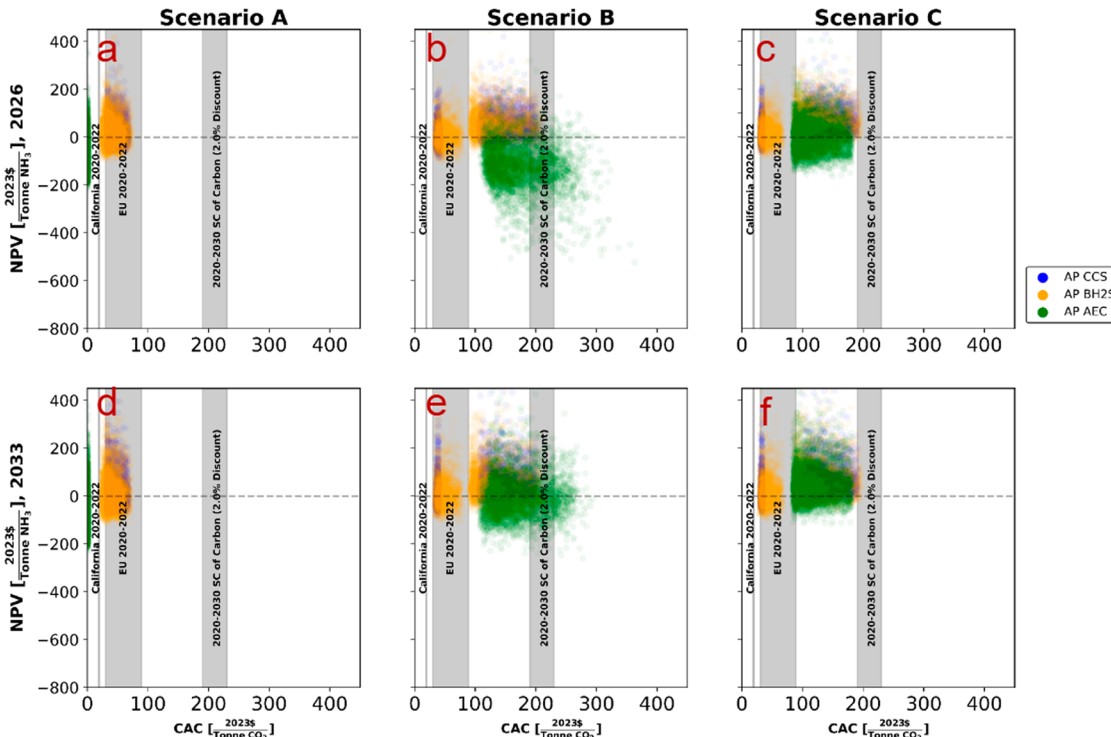

**Fig. 3 | Net Present Value (NPV) versus Carbon Abatement Cost (CAC) for ammonia technologies. a–c** depict Scenario A utilizing grid electricity, Scenario B with colocated wind and solar farm, and Scenario C involving a power purchase agreement with a wind and solar farm for the year 2026, respectively. **d–f** represent the same scenarios projected for the year 2033. We compare the NPV and CAC of various low-carbon ammonia production technologies across three scenarios: Scenario A with grid emissions and lower policy support, Scenario B with an in-situ wind and solar farm resulting in the highest carbon abatement cost, and Scenario C where the ammonia plant enters into a power purchase agreement with a wind and solar farm, receiving only ammonia plant tax credits. Symbols used: blue data points represent the CACs of carbon capture pathways, yellow data points represent biomass gasification pathways, and green data points represent alkaline electrolysis pathways. Additionally, tax quantities from the International Energy Agency (IEA) and California tax prices from the U.S. Energy Information Administration (EIA) are included as reference bands.

programs (See Supplementary Methods), minimizing potentially high "transaction costs" for investors to convert tax credits to cash-equivalent financial support. These transaction costs directly affect the economics of low carbon ammonia production (Fig. 4), as the cash-equivalent support investors receive is lower than the government's support. Most existing studies[9,10] on low-carbon hydrogen and ammonia use potential support in calculating the levelized costs, ignoring those transaction costs.

Thus, considering tax credit at its nominal value in calculations of levelized costs underestimates them—in our case, by 6–12%. While the highest support level under the 45V credit program is $3 per kgH2, the actual support cost of AEC, under the monthly matching scenario, is $4.4 per kgH2 in 2026 and $3.75 per kgH2 in 2033 (Fig. 4, scenario B). Moving to an hourly matching rule will increase AEC's total policy support to $9.6 per kgH2 in 2026 and to $6.8 per kgH2 in 2033. Thus, this additional support (the difference between actual and the $3 per kgH2) is attributed to supporting the hybrid renewable farm, and this additional policy support depends crucially on matching rules. An interesting insight is AEC's cash-equivalent tax credit values under different discount rates in scenario B. At 9%, the cash-equivalent cost of supporting AEC is higher than under a lower discount rate of 2% due to disproportionately high capex, and hence investment tax credits ITC (48E) that investors receive upfront.

### Decarbonizing ammonia production with a policy mix

Carbon is emitted when producing ammonia using SMR, causing global warming. Efficient economic policy to deal with this issue should ensure that agents bear the costs of their actions, leading to the Polluter Pays Principle[11]. This principle is at the heart of many

environmental and economic policies, such as pricing carbon emissions via a market-based emissions cap and permit trading system like the EU Emissions Trading System (ETS). By putting a price on emissions, polluters are penalized, making low carbon ammonia production technologies cost-competitive against incumbent, more emissive technologies (e.g., SMR).

However, implementing this first-best economic policy is not always possible due to political economy considerations[12–15]. Since low carbon ammonia production technologies are not economically competitive with SMR (Fig. 2), a policy option is for the government to provide financial support through subsidies (e.g., IRA's tax credit programs). This part explores the interactions between the IRA's subsidy programs and the EU's carbon pricing instrument. The EU agreed to phase in the Carbon Border Adjustment Mechanism (CBAM) from 2027 with full effect in 2034, and the scope is widened to include hydrogen and ammonia. Figure 5 outlines the economic assessment of these interactions, assuming that US AP targets the EU export market.

Perhaps not surprisingly, we found that under the IRA subsidies and the CBAM policies, low carbon ammonia production technologies outperform conventional SMR even when CCS and BH2S are connected to the US power grid (Scenario A). However, the relative economics of AEC is unchanged—it is still underperforming relative to CCS and BH2S. CBAM increases the attractiveness of AEC (relative to SMR) by a factor of two, increasing AEC's profitability by 37% while penalizing SMR's profit by 14%.

### Renewable hydrogen production rules for low-carbon ammonia

Unsteady state operation of the HB process negatively affects its performance[16], and it is traditionally designed to operate at a steady

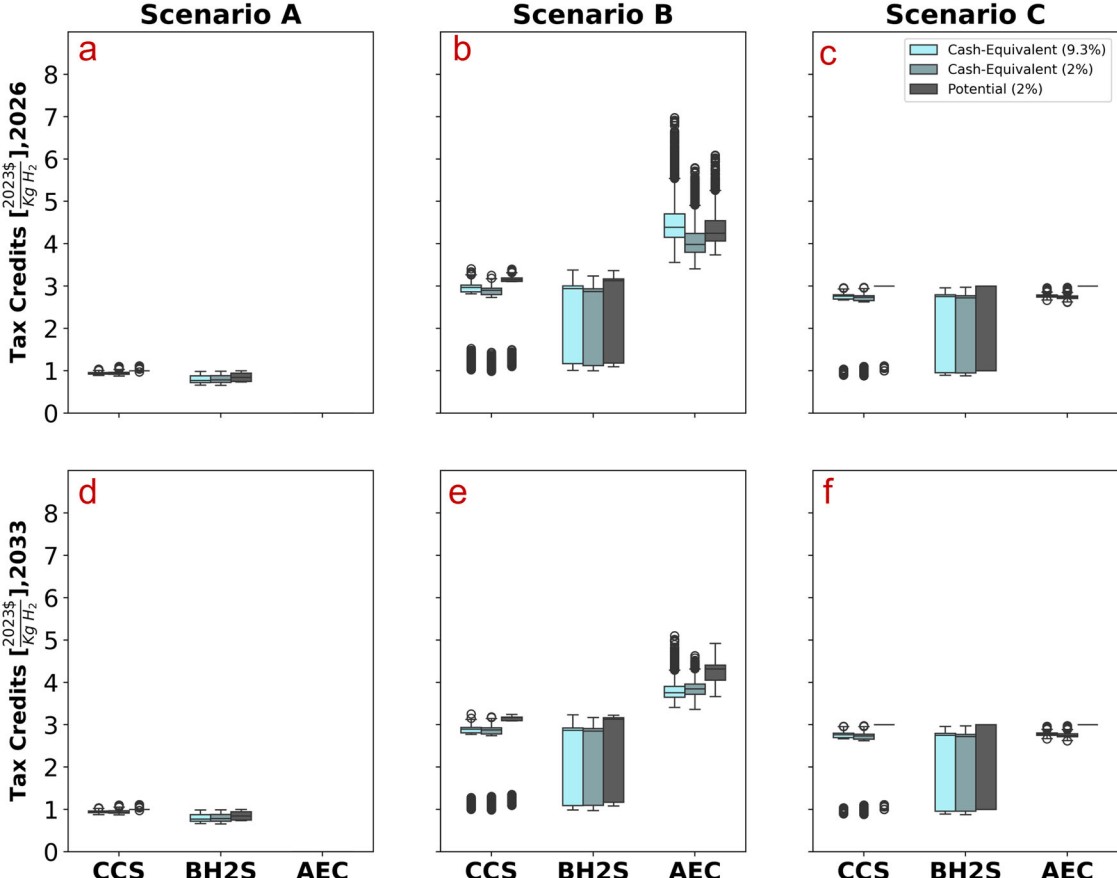

**Fig. 4 | Total policy support for ammonia technologies. a–c** depict Scenario A utilizing grid electricity, Scenario B with colocated wind and solar farm, and Scenario C involving a power purchase agreement with a wind and solar farm for the year 2026, respectively. **d–f** represent the same scenarios projected for the year 2033. We compare the total policy support for various low-carbon ammonia production technologies across three scenarios: Scenario A with grid emissions and lower policy support, Scenario B with colocated wind and solar farm resulting in the highest carbon abatement cost, and Scenario C where the ammonia plant enters into a power purchase agreement with a wind and solar farm, receiving only ammonia plant tax credits without additional wind farm credits. Symbols used: percentage values in the key represent discount rates, potential and cash-equivalent credits are plotted at the same discount rate for comparison, and the cash equivalent at 9.3% represents the weighted-average cost of capital (WACC) discount rate, illustrating the present value of the tax credits from the perspective of the private ammonia plant investor. Similar figures of the hourly and yearly electricity-matched scenarios can be found in the Supplementary Figures.

state[17,18]. Our analysis so far is based on optimal sizing of the hybrid renewable farm capacity such that total monthly electricity generation is matched with electricity requirements by low carbon ammonia production (i.e., monthly matching baseline scenario).

There is considerable debate regarding the matching rule's time granularity (yearly, monthly, or hourly). The time granularity is linked to consequential carbon emissions if low carbon ammonia production is connected to the power grid dominated by fossil fuel generation[19–23]. The IRS, in its final ruling published in January 2025, confirmed that enforcing hourly matching will begin in 2030. This policy is expected to significantly impact the economics of low carbon ammonia production technologies[24]). Thus, we explore the matching rule's time granularity on low carbon ammonia production's economics (Fig. 6) and CAC (Fig. 7).

Moving from monthly to yearly matching only significantly improves AEC's relative (to SMR) economics in 2033 (its relative NPV is 70% higher in the yearly matching case than in the monthly matching). Thus, a looser rule (yearly matching) does not improve AEC's economic performance in the near term (2026). This finding suggests that starting with the monthly matching requirement will not alter the AEC's relative economics while at the same time ensuring lower consequential carbon emissions than the yearly matching rule. Not surprisingly, the hourly rule will only penalize AEC's economic

performance relative to less strict matching rules due to its high electricity requirements while increasing its CAC relative to other low carbon ammonia production options and significantly beyond the social cost of carbon (Fig. 7, Scenario B, CAC of AEC ranges $126-7475 per tCO2e, with a median estimate of $248-324 per tCO2e). Overall, AEC's economic performance is unchanged relative to CCS and BH2S under alternative matching rules. The economics of CCS and BH2S marginally depend on these matching rules because electricity, although necessary, is not the primary energy source for these two low carbon ammonia production pathways.

## Discussion

This study extends existing research on low carbon ammonia production's techno-economic analysis and policy support for low-carbon hydrogen (refer to the Supplementary Discussion). It builds upon our stochastic economic AP model to thoroughly examine key low carbon ammonia production pathways under various policy frameworks. These include subsidies (IRA tax credit programs and the transaction costs of the US tax credit market), carbon pricing policies (EU ETS and CBAM), and low-carbon hydrogen production rules (renewable electricity and hydrogen production matching rules). Our detailed modeling results lead to the following conclusions.

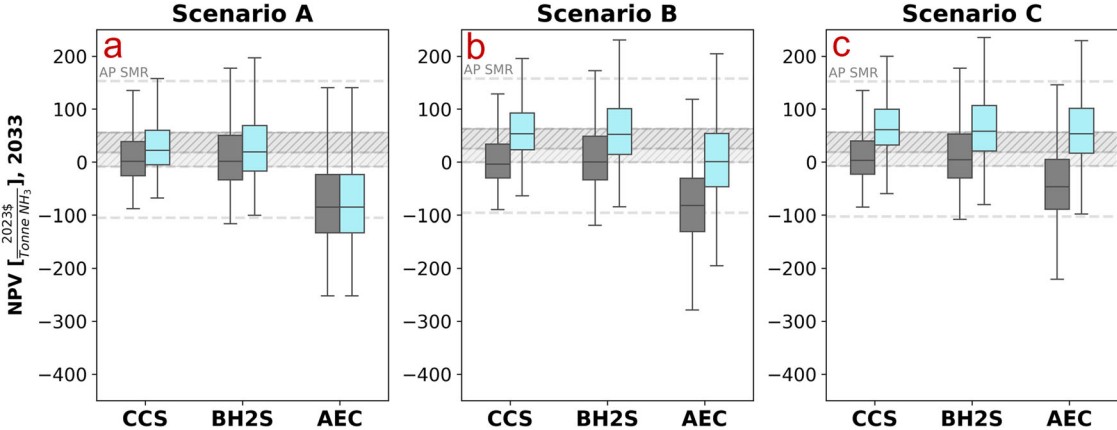

**Fig. 5 | Impact of the Carbon Border Adjustment Mechanism on ammonia technologies. a–c** depict Scenario A utilizing grid electricity, Scenario B with colocated wind and solar farms, and Scenario C involving a power purchase agreement with a wind and solar farm for the year 2033, respectively. We compare the net present value (NPV) of various low-carbon ammonia production technologies against a control scenario with no Inflation Reduction Act (IRA) policy support and ammonia production through steam-methane reforming (AP SMR). Scenario B involves colocated wind and solar farm directly integrated with the ammonia plant, allowing for direct control and optimization of renewable energy generation, whereas Scenario C involves a power purchase agreement (PPA) with external wind and solar farms. Symbols used: light blue data points represent NPVs with the IRA, and gray data points represent NPVs without the IRA. AP SMR refers to Ammonia Plant using Steam Methane Reforming, AP CCS to Ammonia Plant with Carbon Capture and Storage, AP BH2S to Ammonia Plant using Biomass Gasification coupled with Steam Methane Reforming, and AP AEC to Ammonia Plant using Alkaline Electrolysis.

Under the IRA policy framework, both CCS and BH2S prove competitive against SMR, with similar performance levels. The CCS option benefits from a natural market hedge, as ammonia prices closely correlate with natural gas prices. In contrast, BH2S carries a higher market risk since ammonia prices do not correlate with bioenergy costs (refer to the Supplementary Results). BH2S's competitiveness is limited by the costs and availability of relatively inexpensive biomass feedstock. Similarly, the economics of CCS rely on the efficiency of its downstream value chain, including $CO_2$ transport and storage. Risks or perceptions of $CO_2$ leakage could adversely affect the IRA's cash-equivalent support, influencing investment decisions and the adoption rate of CCS technology.

Our analysis assumes a uniform market value for tax credits across all technologies. This assumption is especially relevant for CCS, where environmental and financial risks associated with CO2 leakage pose substantial challenges[25,26]. As per Norton Rose Fulbright, the financial liability for recapture under 45Q credits extends for 15 years[27], increasing the risk for tax equity investors. Consequently, the actual market value of these credits might be significantly lower than their face value, affecting investment decisions and technology adoption rates[28]. Therefore, the presumed effectiveness of tax credits in promoting AP CCS technologies could be more overestimated than other technologies, underscoring the need for a more thorough analysis than what we present here.

AP BH2S, despite its economic advantages over other low carbon ammonia production options, has limitations that are not captured in our model. Its reliance on a large, local biomass reserve can be a constraint if the reserve depletes, reducing scalability or may lead to extra costs for in-situ biomass cultivation. This approach raises concerns about significant land use, possibly better suited for agriculture[29,30]. For instance, cultivating algal biomass in ponds costs about $694–864 per MT, whereas our model considers woody biomass costs $50–118 per MT[31]. Furthermore, Cardoso et al. point to biomass availability as a critical factor in maintaining steady production of ammonia, which they find to be the most important economic factor[32].

AEC has the lowest rate of return among the competing low carbon ammonia production technologies and is not competitive (against SMR) in the near term despite potentially receiving the highest public support. Under IRA, AEC's economics depends on access to a well-developed, cheap, renewable PPA market with 24/7 clean energy matching, which is still under development and costly[33–35]. CCS and BH2S also depend on the PPA market, but their economics are less sensitive to the matching requirements. However, non-linear cost reductions may help AEC in the long term as renewable resource improves (e.g., wind turbine hub height increase) and costs reduction[36,37].

Our findings show that between 2026 and 2033, AEC's economics improve notably due to technological improvements and cost reductions. However, these cost reductions depend on investor participation in early deployment to drive these costs down. If the policy concerns early AEC deployment to drive costs down, IRA subsidies may need to be increased to account for these dynamics (i.e., the $3 per kgH2 tranche increased to $4.8 per kg). While public attention has been focused on the trade-off between the stringency of carbon accounting of the AEC pathway and its early deployment, irrespective of these cost reductions, AEC still underperforms relative to CCS and BH2S in the IRA policy timeline. Thus, technology neutrality in designing policy support for low-carbon technologies is essential. At the same time, the focus should be on stimulating innovation in low-carbon hydrogen technologies and, crucially, their supply chains and market organizations, such as the 24/7 clean PPA market.

The IRA provides unprecedented support for AEC, but the technology underperforms from private and public perspectives: its NPV is lower than those of CCS and BH2S, while its CAC, in some cases, exceeds the social cost of carbon and that of CCS and BH2S. Although, in some cases, exceeding the recent EU carbon prices, IRA subsidy programs are cost-effective in terms of value for public money in supporting hydrogen-based climate mitigation technologies.

This research underscores the importance of considering nuances of the US tax credit markets because tax credits under the IRA will not translate into actual subsidies on a parity level. Thus, the levelized cost approach should explicitly consider these transaction costs. Ignoring the complexity of the tax credit market and its interactions with the PPA markets will result in an underestimation of low carbon ammonia production levelized cost, especially those with significant barriers to deployment and demonstrate their efficiency at scale[38–40]. Risky and

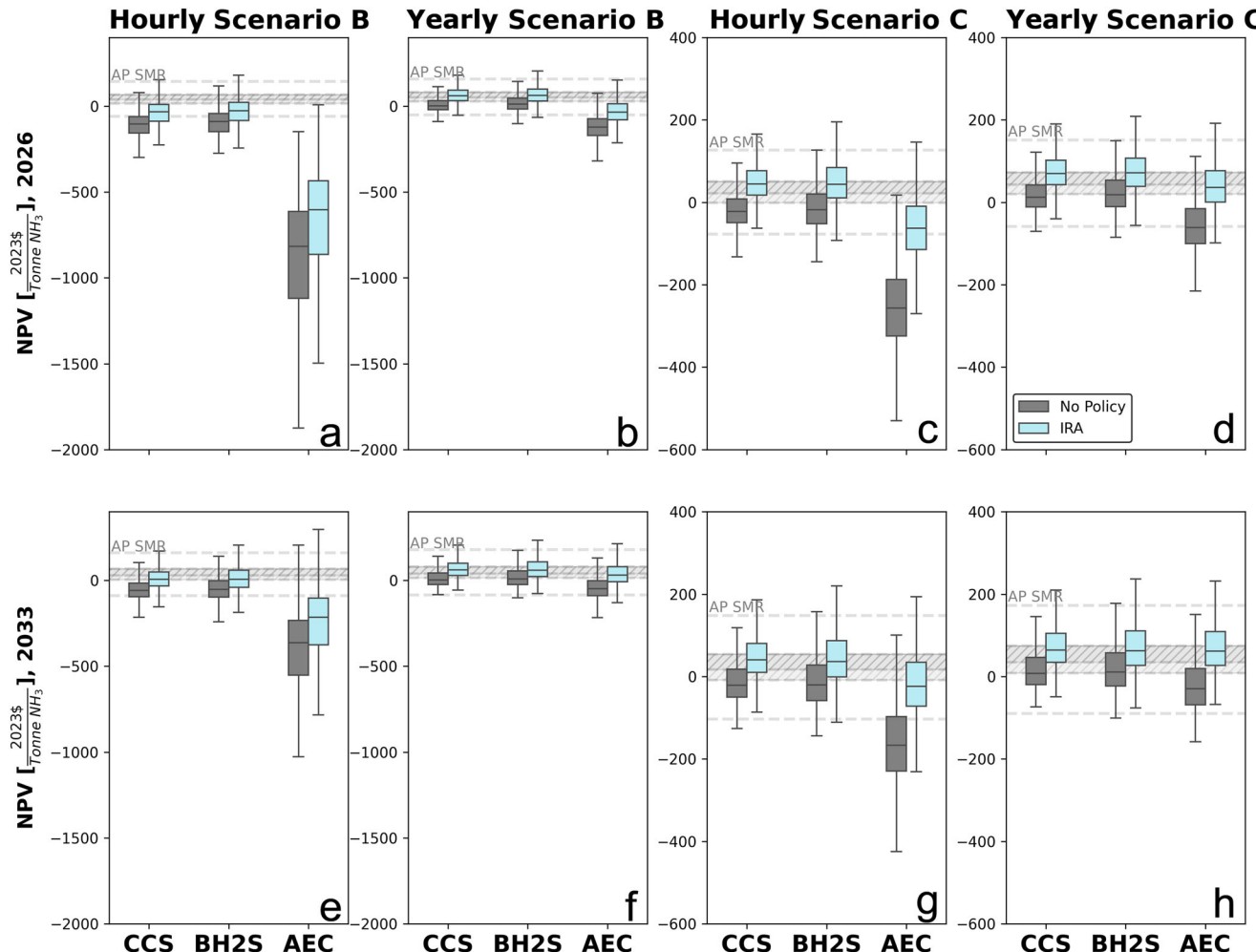

**Fig. 6 | Net present value of ammonia technologies: hourly and yearly matching scenarios. a, b** depict Scenario B utilizing hourly and yearly matching, respectively, while (**c**) and (**d**) depict Scenario C utilizing hourly and yearly matching for the year 2026. Panels (**e**) and (**f**) represent Scenario B with hourly and yearly matching for the year 2033, respectively, and (**g**) and (**h**) represent Scenario C with hourly and yearly matching for the year 2033, respectively. We compare the net present value (NPV) of various low-carbon ammonia production technologies against a control

scenario with no Inflation Reduction Act (IRA) policy support and ammonia production through steam-methane reforming (AP SMR). Symbols used: light blue data points represent NPVs with the IRA, gray data points represent NPVs without the IRA, and gray lines represent AP SMR. The y-axis ranges vary across panels to accommodate the full range of NPVs for each technology, with AP AEC exhibiting a broad range of negative NPVs.

unproven (at scale) technologies (AEC has the highest risk profile) will involve higher capital costs and verification, compliance, and monitoring costs, potentially significantly increasing transaction costs beyond what this study assumes. Technologies with high-risk profiles will be costlier for the government to support, implying that the government may consider underwriting risks to lower capital costs for investors and, hence, lower support costs per unit of H2 (e.g., by 7–21% for AEC if its WACC is reduced from 9% to 2%, Fig. 4).

In the foreseeable future, there is little chance of putting a price on carbon emissions in the US. Instead, the IRA framework offers unprecedented financial incentives to stimulate private capital into low-carbon energy technologies. On the contrary, the EU's flagship carbon pricing is regarded as the first-best economic policy to address carbon emissions[41–45]. Perhaps not by design, the interactions between the CBAM and IRA will likely mean stronger incentives to decarbonize US AP than standalone IRA. The relatively small carbon taxing and the opportunity to trade CBAM certificates could substantially increase the relative economics of US-based low carbon ammonia production: grid connection (Scenario A) is now a cost-effective option for at least CCS and BH2S (their NPVs are higher than those of SMR). Under CBAM and IRA, the CAC to decarbonize US AP via CCS and BH2S could be

much lower (23-73 $ per tCO2e), falling in the range of recent EU carbon prices (Fig. 3: Scenario A). This finding reconfirms the potential effectiveness of multiple policy instruments in a "second-best" world[44,46–48] to reduce the US AP carbon emissions.

There is considerable debate about consequential emissions from renewable electricity and hydrogen production matching rules. Starting with the monthly matching rule will not unduly penalize AEC's economics while ensuring lower consequential emissions than the yearly rule. While the hourly matching rule ensures limited consequential emissions from the AEC, its unfavorable economics will unlikely stimulate private investment. Hourly-matched AEC pathway seems unlikely a worthwhile avenue to pursue from the public policy perspective because its support cost outweighs the carbon savings benefits (in most cases, AEC's CAC is substantially higher than the social cost of carbon). The costs of supporting an hourly-matched AEC pathway make it a less appealing option from a public policy standpoint, as AEC's carbon abatement cost is often much higher than the social cost of carbon. The high cost stems from considerable investments in renewable and energy storage capacity to ensure stable electrical input as the Haber-Bosch process requires a steady state mode of operation (see the Supplementary Discussion). We consider

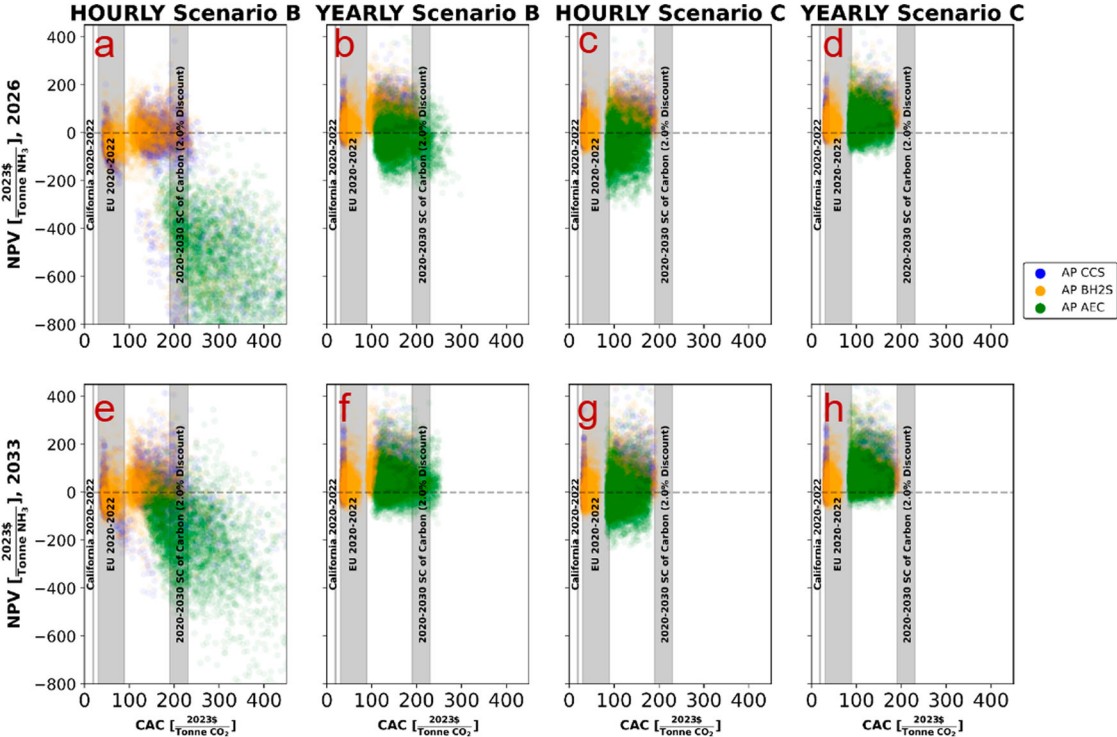

**Fig. 7 | Net present value versus carbon abatement cost for ammonia technologies.** (**a**, **b**) depict Scenario B utilizing hourly and yearly matching, respectively; (**c**) and (**d**) depict Scenario C utilizing hourly and yearly matching for the year 2026, respectively; (**e**) and (**f**) depict Scenario B utilizing hourly and yearly matching for the year 2033, respectively; and (**g**) and (**h**) depict Scenario C utilizing hourly and yearly matching for the year 2033, respectively. We compare the net present value (NPV) and carbon abatement cost (CAC) of various low-carbon ammonia production technologies under hourly and yearly matching scenarios. Symbols used: blue data points represent the CACs of carbon capture pathways, yellow data points represent biomass gasification pathways, and green data points represent alkaline electrolysis pathways. The y-axis ranges vary across panels to display the full bar chart for each technology.

our hourly matched results to be the absolute worst-case scenario with no flexibility.

AP is expected to almost triple (688 Mt per year) by 2050, with 83% from renewable ammonia (IRENA, 2022). If renewable ammonia is part of this vision, then advancements in the flexibility of the HB process are a crucial avenue for research and development. Some research has highlighted the challenges of flexible HB. The literature reports HB may handle wide ranges of output (5–80% of capacity) and ramping rates (20% capacity per hour) based on feasibility studies and industry opinion[16,49,50]. However, the demonstration of flexible HB at a small scale is only starting, while the additional costs of flexible HB loops at a commercial scale are unclear[51,52]. Given the current industry state versus the optimistic techno-economic literature, it may be a reality that flexible HB may exist commercially in the next ten years but beyond the IRA timeline. Nevertheless, the incentives for making flexible HB are clear under an electric grid with increasingly fluctuating renewables: our results highlight that the economic benefit of flexible HB could be substantial: $2.0-2.2 bn or $53-60 per tNH3.

To conclude, this research finds that to decarbonize the AP, there are critical areas for policymakers and the academic community to focus on in the next decade: (i) adapting HB to variable bioenergy quality and process efficiency while ensuring feedstock's sustainability and availability (ii) ensuring safe transport and permanent storage of CO2 while de-risking the CCS value chain, (iii) supporting research and development to drive down cost and efficiency improvements of flexible HB, renewable energy, and electrical and hydrogen-based storage, (iv) developing a robust policy support framework that ensures technology neutrality and competition while recognizing the

nature of "dynamic" technology cost reduction[53] as well as interactions between policy instruments and between technologies.

## Methods

This section offers an outline of our AP economic model. A comprehensive presentation of the structure and constitutive components of the proposed methodological framework is included in the supplemental methods. We describe the stochastic discounted cash flow model and give an overview of policy modeling, scenarios considered in this study, and key assumptions. All related chemical processes, techno-economic parameters, and assumptions are listed and described in the accompanying supplementary information. Figure 8 depicts the systematic approach to every technology option in the present study. Table 1 describes the various technologies along with their IRA policy eligibility.

### Baseline Process Economic Model

To address the impact of the IRA on the low-carbon AP pathways, we first developed a fixed capital investment and operational expenditure AP model, utilizing the techno-economic performance assessment framework proposed by Peters and Timmerhaus[54]—details can be found in the the supplementary methods section. Figure 2 below illustrates a simple description of the model. The implementation of this probabilistic framework was done using Python's NumPy libraries on random number generators[55].

### Stochastic Discounted Cash Flow Model

This section presents our stochastic discounted cash flow model. We first define the basic deterministic discounted cash flow and then

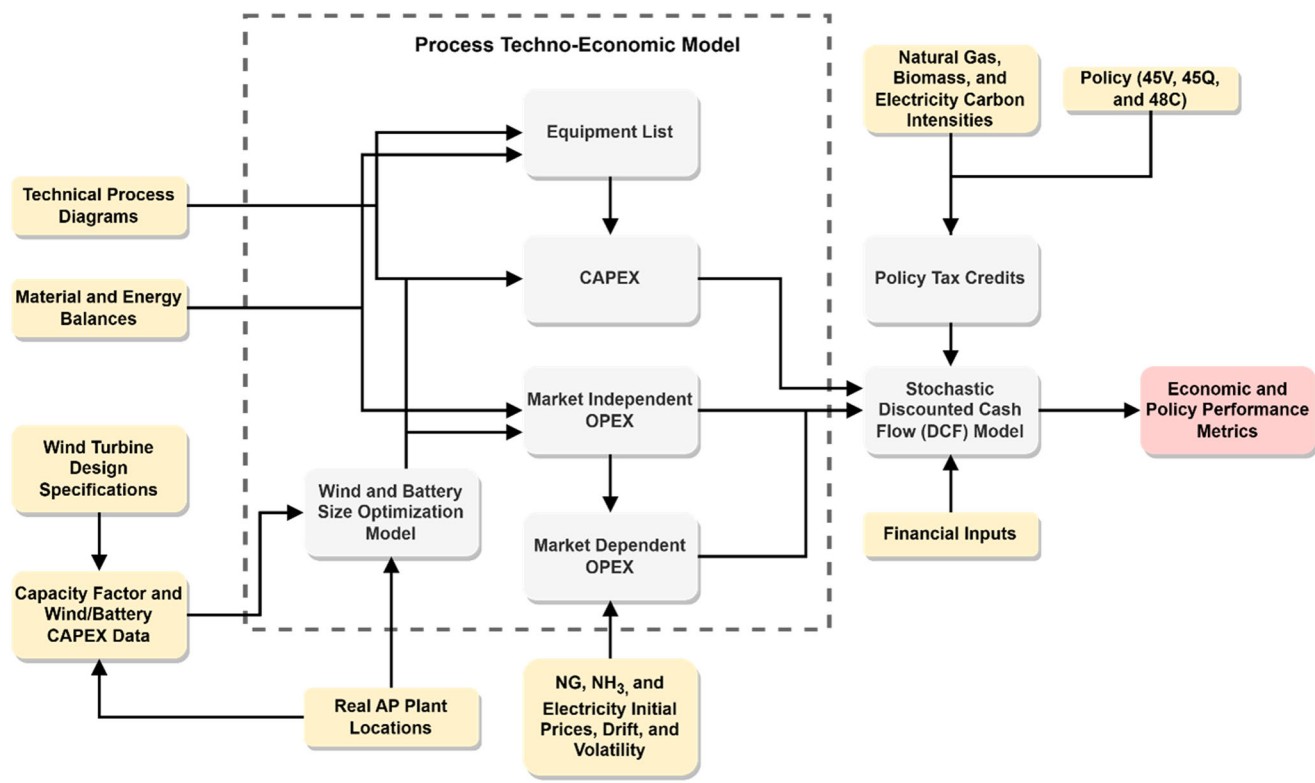

**Fig. 8 | Model schematic.** This figure provides a simplified representation of the research methodology. Light yellow boxes represent model inputs, white boxes denote computational blocks, and the red box indicates the key model outputs.

### Table 1 | Summary of analyzed technologies for AP and their policy eligibility

| Label | Technology | Feedstock[a] | IRA Credit Qualification |
|---|---|---|---|
| AP SMR | Conventional Steam Methane Reforming (SMR) | Natural gas, air | None |
| AP SMR CCS | SMR + Carbon Capture System (CCS) | Natural gas, air | Max [45V, 45Q] and Max [45Y, 48E] |
| AP BHS | Indirect Biomass Gasification + SMR (BHS) | Biomass, air | 45V and Max [45Y, 48E] |
| AP AEC | Alkaline Electrolysis Cell (AEC) | Water, air | 45V and Max [45Y, 48E] |

[a] To produce $NH_3$, a nitrogen and hydrogen source is required. Air is fed to an air separation unit (ASU) for all cases to obtain high-throughput, high-purity $N_2$ gas. *AP SMR* Ammonia Plant using Steam Methane Reforming, *CCS* Carbon Capture System, *BHS* Biomass Gasification coupled with Steam Methane Reforming, *AEC* Alkaline Electrolysis Cell, *IRA* Inflation Reduction Act.

outline critical inputs considered in our stochastic discounted cash flow model. Lastly, we summarize the policy and scenarios modeled.

The economic performance of each technology pathway is assessed using a Net Present Value (NPV) per lifetime ammonia produced. NPV is the sum of the present value of all cash flows at each period (monthly basis) over the lifetime amount of ammonia production, $M_{NH3}$ as shown by equation (1):

$$NPV_j = \frac{1}{M_{NH3}} \sum_{T=0}^{L} \frac{CF_{j(T)}}{[(1+i)(1+d)]^T} \quad (1)$$

where $M_{NH3}$ is the lifetime amount of ammonia produced, $CF(T)$ is the cash flow at time $T$, and $d$ and $i$ are the real discount and inflation rates, respectively. The cash flow $CF_j(T)$ equals the sum of seven cost components as shown below in equation (2):

$$CF_j = FCI_j(T) + Land_j(T) + WC_j(T) + PMT_j(T) + Sales_j(T) + OPEX_j(T) \\ + Tax_j(T) + Credits_{CE_j}(T)$$

$$(2)$$

where $FCI_j(T)$ is the fixed capital investment, invested over the first three years and derived in the supplementary methods; $Land_j(T)$ and

$WC_j(T)$ are the costs of purchasing land and injecting working capital to begin operation; $PMT_j(T)$ represents the payment of borrowed capital for plant construction; $Sales_j(T)$ and $OPEX_j(T)$ represent revenue from selling ammonia and the plant's operational costs, respectively; $Tax_j(T)$ is the income tax and $Credits_{CE_j}(T)$ represents the cash-equivalent IRA tax credits, described in the policy section below. Market prices of ammonia, natural gas, and electricity are modeled with Geometric Brownian Motion (GBM) (see supplementary methods).

Traditional economic valuation methods using discounted cash flow models do not explicitly incorporate and quantify uncertain conditions that could impact economic performance due to model nonlinearities and constraints. Evaluating economic performance under average conditions may not represent true average performance, leading to the "flaw of averages" as described in probability theory[56]. This limitation can lead to erroneous investment decisions and comparative assessments of projects.

Monte Carlo simulation techniques address these limitations by offering probabilistically unbiased estimates of the expected NPV and additional valuable statistical measures such as standard deviation and Value at Risk/Value at Opportunity. This approach accommodates multiple sources of uncertainty, unlike traditional sensitivity analysis, which varies only one model input at a time. Our methodology

quantifies uncertainties associated with value-driving model input variables, including market parameters (NG, NH3, and electricity), policy criteria (48E credits), and lifecycle CO2 intensities (CI). Financial inputs such as equity, cost of debt, return on equity, loan terms, depreciation, and income taxes are kept constant. Market parameters are simulated using general Brownian motion, calibrated for AEO 2023 and  scenario A.

CI can vary significantly due to uncertain value chain emissions. We divide CI into SMR emissions, NG upstream lifecycle emissions, biomass upstream emissions, and grid electricity lifecycle emissions. SMR CI is governed by the mass and energy balances of the AP plant and originates from Lewis et al. (2022). NG upstream and grid electricity lifecycle emissions are uncertain due to value chain emissions and are derived from Nicholson & Heath[57]. NG and grid electricity CIs vary depending on the supply chain and grid energy composition. For instance, a grid composed of wind and solar energy may be net-zero, while a coal-based grid incurs significant emissions penalties.

Our analysis excludes the lifecycle emissions of water and construction materials. Water has negligible carbon intensity, and construction material emissions are difficult to calculate without a detailed plant inventory[58]. For a comprehensive description of input variables, please see supplemental methods section.

## Policy modeling and key assumptions

The AP technologies that we consider in this paper will be eligible for IRA programs 45V, 45Q, 45Y, and 48E, which consider tax credits dependent on material production (production tax credits, PTC, and CO2 sequestration credits, CSC) as well as CAPEX-dependent tax credits (investment tax credit, ITC). Table 2 describes these policy programs.

Credit 45V depends on the ammonia plants' lifecycle carbon intensity (CIs). Credit 45Q depends on the difference in the direct emissions between the SMR and CCS plants.

CI values vary based on the supply chain of the utilities, feedstocks, and construction materials. Using the results of lifecycle assessments of natural gas, electricity, and biomass, we estimate a range of CI for each technology to quantify supply-chain emissions intensity and life cycle scope uncertainties[57]. AP AEC and AP BH2S are eligible for 45V and 45Y or 48E (when AEC and BH2S own low-carbon electricity generation facilities). AP CCS is eligible for 45V or 45Q and 45Y or 48E (when CCS owns low-carbon electricity generation facilities).

Given the policy incentives and CIs, the cash-equivalent tax credits the ammonia plants will receive can be modeled through equations ((3)-(5)) as follows:

$$Credits_j(T) = \max\left[45V \cdot \dot{m}_{H_2}, 45Q \cdot \dot{m}_{H_2} \cdot (CI_{SMR} - CI_j)\right]$$
$$+ \max\left[CAPEX_{\text{wind farm},j} \cdot 48E, (E_{AP} + E_{market}) \cdot 45Y\right] \quad (3)$$

$$Credits_{CE_j}(T \parallel Tax_j(T) | > Credits_j(T)) = Credits_j(T) \quad (4)$$

$$Credits_{CE_j}(T \parallel Tax_j(T) | < Credits_j(T)) = -Tax_j(T) + F_t{}^*(Credits_j(T) + Tax_j(T)) \quad (5)$$

where $Credits_j(T)$ is the amount of tax credits the plant j legally qualifies for, and $Credits_{CE_j}$ is the cash-equivalent tax credit which takes transaction costs from "transferability" into account (see Table 2). 45V is a piecewise function of CI - corresponding to the values in Table 2, and has a lifetime of 10 years from the start of plant operation. 45Q is a constant, $85 per tCO2, and non-zero for the first ten years of operation. 48E is the ITC factor applied to the CAPEX of the wind farm and battery facility - where the CAPEX is a function of the nameplate capacity of the wind and battery storage (see scenarios below). 45Y is the PTC associated with the wind farm, which is mutually exclusive with 48E and is a function of the sum of the electricity supply to the AP plant ($E_{AP}$) and excess electricity ($E_{market}$) sold to third parties. $\dot{m}_{H2}$ is the monthly flowrate of H2. $F_t$ is an exchange rate of USD per tax credit (see supplementary table 19). We assume that the excess tax credits are sold to third parties for cents on the dollar after the initial "direct pay" period of five years (see supplementary table 19).

The cash-equivalent tax credits (equations ((4)-(5))) are not always equal to the nominal tax credits because there are cases where there is not enough income tax to abate with the tax credits. In these cases, we capture this effect by abating the entirety of the income tax and adding the remaining tax credits at a fraction of their value to quantify the transaction costs of "transferability". When the tax credits are less than the income tax, the cash equivalent tax credits equal the total credits. We assume all tax credits, except for 45Y, qualify for "direct pay" for the first five years of operation[59].

## Baseline scenarios

The CI of electricity inputs is important for low-carbon AP, especially the AEC technology option, to be qualified for policy support. We, therefore, consider three scenarios of how an AP plant can source its electricity demand (see supplementary table 32).

For scenarios B and C, we consider IRA's subsidies and how they impact low-carbon AP economics when combined with upfront investments in wind and battery (scenario B) or locking into a long-term PPA with a renewable developer (scenario C). Scenario B represents a business model whereby a low-carbon AP plant builds and owns renewable generation to feed into AP production. It is an integrated business model whereby the AP producer invests in a hybrid renewable farm with a battery facility near the main AP plant. In this case, in addition to other IRA policy supports, the AP plant owner can claim

**Table 2 | Summary of the Inflation Reduction Act's programs considered for this analysis[6]**

| Policy Program | Description and Benefits |
| --- | --- |
| **New Clean Hydrogen Production Tax Credit (45V)** | An intensive 10-year tax credit for clean hydrogen production of varying magnitude based on a Well-to-Gate Lifecycle Emissions intensity measured in $\frac{KgCO_2eq}{KgH_2}$. The tax credit values are as follows: 3.00 per Kg $H_2$ for CI between 0-0.45, 1.00 per Kg $H_2$ for CI between 0.45-1.5, 0.75 per Kg $H_2$ for CI between 1.5-2.5, and 0.60 per Kg $H_2$ for CI between 2.5-4.0. |
| **Carbon Capture and Sequestration Tax Credit (45Q)** | Imposes an intensive 12-year carbon capture tax credit on carbon capture facilities. The tax credit is $85 per MT CO2. To be eligible, the plant must emit more than 12,500 metric tons of CO2 per year at the baseline. |
| **New Clean Electricity Investment Tax Credit (48E)** | Provides an investment tax credit for wind and solar farm and battery storage projects.[a] 30% of the CAPEX of the wind and battery system is granted as tax credits and expires at the end of 2032. |
| **New Clean Electricity Production Tax Credit (45Y)** | An intensive tax credit aimed at rewarding low-carbon electricity generation. 1.5 cents per kWh for wind and solar projects started before 2025. PTCs expire in 2033 or when 75% emissions reductions are achieved. |

[a]We assume 48E credits also support the cost of battery storage systems.

45Y or 48E tax credits as generation from wind is considered a low-carbon, clean electricity source. The downside of this case is the upfront funding for the wind and battery facilities.

We chose to let the AP SMR baseline not be subject to the conditions of scenarios B and C. Instead, we let the AP SMR benchmark be under 2026/33A conditions as that is the traditional configuration of AP.

On the other hand, scenario C outlines a business model whereby the low-carbon AP plant signs a corporate Power Purchase Agreement (PPA) with a renewable developer. In this case, the renewable developer invests in a new hybrid wind farm and sells this power to the AP plant under a long-term fixed-price PPA. In this case, the AP plant is forgoing the 45Y or 48E tax credit program (as the AP plant is not the renewable farm owner), but it is also avoiding the upfront renewable farm investment cost. Thus, comparing scenarios B and C will show the implication of the two business models on the economics of AP.

In both cases, having a battery helps modulate wind power generation fluctuations. The cost of a battery facility can be considered an opportunity cost of renewable electricity matching to ensure that green hydrogen truly consumes renewable power. In fact, after 2028, the IRS plans to enforce hourly matching[24] - which is consistent with Ricks et al.'s findings[20].

There is a hot debate regarding the time resolution of this matching requirement (e.g., yearly, monthly, or hourly; see the supplementary scenario-specific methods section) because different time resolutions will have a significant impact on carbon emissions (for a detailed discussion of this question see[20]). For the baseline scenarios, we consider the monthly matching of renewable electricity to fuel hydrogen production under the AEC pathway. We offer sensitivity analysis modeling yearly and hourly matching requirements for the AEC pathway (see sensitivity analysis in supplementary results section). Lastly, scenarios B and C also meet the deliverability and additionality requirements (the hydrogen production facility is directly connected to a new hybrid renewable farm, offtaking electricity directly from this facility).

We should note that the fuel's LCA assumptions are consistent with the ones we use for feedstock (AP CCS and AP BH2S). Regarding electricity grid CI, the AP plant is assumed to be within a range of locations across the US (see the supplementary methods), so the EIA's 2023 average US industrial electricity mix predictions are used for scenario A[60] (See the Supplementary Figs. section for electricity prices in the model). We utilize the wind and battery CAPEX predictions by Bistline et al. for scenarios B and C and solar predictions from EIA's AEO 2022[61].

Because time dimensions are important (US power grid decarbonization pace and cost reduction in low carbon energy technologies), we are conducting this analysis assuming AP's start of operations in 2026 and 2033. We will refer to these scenarios by the year when operations begin. For example, scenario A in 2026 will be 2026A. Thus, comparing results for 2026 and 2033 will reveal the implications of cost and grid CI reduction on the economics of low-carbon AP. For technologies in 2033, we assume the equity share of the FCI appreciates at the risk-free rate of 4.25 percent[62].

## Sensitivity Analyses

We analyze two sets of sensitivities critical to the economics of low carbon AP under the IRA framework - a potential impact of the EU's Carbon Border Adjustment Mechanism (CBAM) and the implications of matching renewable generation with hydrogen production.

The EU is the third largest importer of US-based ammonia production in 2021[1]. The EU agreed to phase in CBAM from 2027, and the scope is widened to include hydrogen and ammonia. Therefore, we also model a CBAM sensitivity scenario in which ammonia exports to the EU are subject to its carbon tax. Thus, the baseline scenarios are analyzed with and without an EU carbon tax mechanism. The objective is to gauge the impact of IRA and CBAM policies on the economics of US-based AP.

We model this sensitivity scenario by implementing a carbon tax. The European cap is set at the average CI of the top 10% least emissive AP production facilities[63] - which starts at 8.82 Kg $CO_2e$ per Kg $H_2$ and decreases by 1.4% per year as per European Commission and literature estimates[63–65]. Any additional emissions above the cap must be penalized by an equivalent purchase of CBAM certificates. We price the CBAM certificates as uniformly distributed between $35-100 per $tCO_2$ - corresponding to the latest EU ETS $CO_2$ prices[63].

Thus, when the CI difference between the EU cap and the US AP technology is negative (meaning the US AP technology is more emissive), the US AP plant incurs a cost, decreasing its cash flow in proportion to the magnitude of the CI difference and CBAM certificate price. In contrast, when the margin between the EU cap and US technology is positive, it leads to a boosted cash flow from the surplus CBAM certificate sales. Thus, we implicitly assume here that low-carbon AP plant can sell surplus CBAM certificates to other carbon-intensive AP exporters who want to sell their products in the EU; that is, there is potentially a trading scheme for CBAM certificates available for AP exporters and that these certificate prices are linked to EU ETS price.

For the second sensitivity analysis, we note the unsteady state of the Haber-Bosch process negatively affects its performance[16]. The Haber-Bosch process is traditionally designed to operate at a steady state. Thus, for this sensitivity case, a battery facility is assumed to be sufficient to ensure a constant output of a renewable facility to fuel the AP plants (See supplementary section).

We explore yearly, monthly, and hourly matching with a hybrid wind farm and find the appropriate facility capacities through an optimization model (See supplementary methods section). The strict hourly matching scenario comes with a relatively high battery cost (due to the sizing of wind, solar and battery facilities). There is a debate regarding whether such a strict hourly matching will impede the roll-out of green hydrogen[19–23]. On the other hand, while economically more advantageous and relaxed, yearly matched green electricity production will incur additional emissions, depending on the grid CI. We explore the impact of yearly and hourly matching on the economic benefits of low-carbon ammonia.

## Model validation

Our plant-level costing model appears credible when benchmarked against data produced by the IEA's Ammonia Technology Roadmap[1]. We created a deterministic version of the current model and tested input and sensitivity parameters that the IEA used to recreate similar levelized costs and uncertainty ranges. We analyze these results in the supplementary results section.

To ensure probabilistic convergence, our stochastic model underwent 4,000 simulation runs, confirmed by a convergence analysis testing the change in model outputs for a given change of additional simulations. More details can be found in supplementary results section and the attached supplementary datasets.

Further validation was achieved through a sensitivity analysis, which examined key input assumptions to assess the model's quality and directional impact based on changes to the input parameters (see supplementary results section). Collectively, these evaluations affirm that the model yields economically sound results.

## Data availability

The data generated in this study are provided in the Supplementary Information file.

## Code availability

The code used in this study is available on GitHub under the user Eduita. https://github.com/Eduita/AP_IRA_Model.

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

## Acknowledgements

We want to express our heartfelt gratitude to William Garvey and Paul Jasmin for their invaluable assistance in developing the literature review on ammonia flexibility. We are deeply thankful to Yihan "Emma" Zhang for her help in developing the optimization algorithm, conducting the literature review, and providing support with the supplemental information.

## Author contributions

E.I. performed the techno-economic modeling, programming, visualizations, formal analysis, writing, and editing. C.K.C. contributed to research conceptualization, electricity optimization modeling, formal analysis, writing, revision, advising, and funding acquisition. N.K. provided revisions and advising. E.I., C.K.C., and N.K. reviewed and approved the final manuscript.

## Competing interests

The authors declare no competing interests.
