## [Peer Review file · Nature Communications]

REVIEWER COMMENTS

Reviewer #1 (Remarks to the Author):

this study considered the key low-carbon ammonia production pathways - steam methane reforming with carbon capture and storage (CCS), biomass gasification (BH₂S), and electrolysis (AEC) -

under multiple policy frameworks – subsidies, carbon pricing, renewable hydrogen rules. This MS can be published after Major revision. My comments to the authors are;

1. Abstract should be improved, I cannot see the objective of this research and methodology, etc.
2. Literature survey should include recent ammonia production research and energy policy researches. Pls expand.
3. you should explain what is your research implication on energy policy? why do you recommend AP? what is the importance of your research?
4. what is your stochastic economic analysis methodology? and how? Pls explain.
5. is your research based on theoretical analysis or actual data analysis framework?
6. What are the climate policy implications on the AP processes?
7. What are the constraints of climate policy design which influenced the policy making and economy framework?
8. why did you select the HB process for ammonia synthesis?
9. Authors should discuss the economy framework and cost.
10. The figures caption should be improved.
11. what is the difference between scenario A,B,and C in fig.3? Figures are not clear.
12. what are the implications of energy and climate policy on AP? Pls discuss.
13. discussion should be expanded, Pls discuss more.

Reviewer #2 (Remarks to the Author):

This work delivers a stochastic economic analysis of the plant-level ammonia production model considering key low-carbon ammonia production pathways such as steam methane reforming w/ CCS, biomass gasification, and electrolysis. The article comprehensively addresses various aspects, including economic analysis, policy implications, and technology considerations, providing a holistic view of the topic. Overall, the paper deals with a pivotal subject, it is very well written, and the figures are appealing, therefore; this reviewer finds no constraints in advising this article for publishing in Nature Communications journal after some minor revisions.

The scenarios (A, B, C) are outlined effectively, but a brief introductory paragraph summarizing their significance could enhance the reader's understanding before delving into the details. Consider using concise bullet points to highlight key features of each scenario.

The explanation of the Monte Carlo simulation approach is commendable. Emphasize the importance of overcoming limitations in traditional models, making it explicit how this methodology enhances the study's robustness.

The sensitivity analyses are well-structured. Please just clearly articulate the rationale behind each sensitivity analysis, emphasizing the potential real-world implications and why they are crucial for understanding the economic dynamics of low-carbon ammonia technologies.

The model validation and robustness section is detailed and provides assurance of the model's credibility. Consider summarizing key validation results in the main text and directing the reader to the supplementary files for more in-depth information.

Emphasize the transparency of assumptions, particularly regarding carbon intensity variations and uncertainty sources.

The following article might also be of interest for your references and research: Cardoso, et al., Small-Scale Biomass Gasification for Green Ammonia Production in Portugal: A Techno-Economic Study, *Energy&Fuels*, 2021, 35, 17, 13847–13862.

Reviewer #3 (Remarks to the Author):

The authors present a stochastic economic assessment of different avenues for ammonia production.

The work is relevant and results are interesting. However, it would be beneficial to clarify some points. The structure of the work is unconventional, where the premises and assumptions are provided after the results.

A list of acronyms would be very helpful

Regarding Monte Carlo simulations, what tools were used?

In NPV calculations, what criteria was followed to set the selling price of product?

Why is alkaline electrolysis preferred to other alternatives such as PEM, which present a high degree of flexibility suitable for intermittency of renewable sources?

A point to discuss is the scale which can be aimed with different primary energy sources. This can also present a relevant competitive advantage for natural gas based SMR in terms of economies of scale.

Regarding the discussion about matching, given that wind presents significant seasonal fluctuations, is it not an argument to consider a yearly span in the evaluation. Please clarify the methodology.

Some insights regarding natural gas costs that would render the renewable route more attractive would be interesting.

Why is the renewable resource only limited to wind? Solar PV presents very attractive costs that can improve competitiveness of AEC.

Reply to the Reviewers' Comments

The authors thank the Editor and anonymous reviewers for their time, insightful remarks, and helpful suggestions, which improved the paper considerably. Every possible effort has been made to address and incorporate all their suggestions and remarks into a carefully prepared revised version. A detailed point-by-point reply to the reviewer's comments immediately follows:

Replies

Reviewer questions are in black font; replies are in blue font; updated text is in orange.

Reviewer #1 (Remarks to the Author):

This study considered the key low-carbon ammonia production pathways - steam methane reforming with carbon capture and storage (CCS), biomass gasification (BH2S), and electrolysis (AEC) - under multiple policy frameworks – subsidies, carbon pricing, renewable hydrogen rules. This MS can be published after Major revision. My comments to the authors are;

1. Abstract should be improved, I cannot see the objective of this research and methodology, etc.

We are grateful for the reviewer's constructive critique regarding the abstract of our manuscript. Upon reflection, we acknowledge that our initial abstract overly emphasised the results while omitting crucial details about the research's key objectives and methodological approach. In response to this feedback, we have revised the abstract to ensure a balanced presentation. The revised version places more emphasis on the primary objectives and methodological foundations. See the revised abstract below:

“This study aims to evaluate the economic feasibility of low-carbon ammonia production (AP) pathways, including steam methane reforming with carbon capture and storage (CCS), biomass gasification (BH2S), and electrolysis (AEC), in the context of various policy frameworks like subsidies, carbon pricing, and renewable hydrogen regulations. Employing a stochastic techno-economic analysis at the plant level, we applied a net present value (NPV) approach to assess these technologies under the Inflation Reduction Act (IRA). Our findings reveal that CCS and BH2S pathways exhibit strong economic potential, mainly due to their cost-effectiveness and minimal public support needs. Conversely, the AEC method faces significant economic hurdles due to its higher costs and lower efficiency. We propose several recommendations for policy and academic focus to efficiently decarbonise AP. These include adapting the Haber-Bosch process to account for variable bioenergy quality, ensuring safe CO₂ transport and storage for CCS, advancing research and development to lower costs and improve efficiency in flexible Haber-Bosch, renewable energy, and storage technologies, as well as creating a technologically neutral policy framework that accounts for dynamic cost reductions and the interplay between policy instruments and technologies.”

The previous abstract:

“Building on the stochastic economic analysis of the plant-level ammonia production (AP) model, this study comprehensively considers key low-carbon AP pathways - steam methane reforming with carbon capture and storage (CCS), biomass gasification (BH2S), and electrolysis (AEC) - under multiple policy frameworks – subsidies, carbon pricing, renewable hydrogen rules. Through a net present value (NPV) approach, CCS and BH2S demonstrate strong

economic potential under the Inflation Reduction Act due to cost-effectiveness and limited public support requirements. In contrast, AEC faces economic challenges due to high costs and low efficiency. To efficiently decarbonise AP, policymakers and academia should prioritise (i) adapting Haber-Bosch (HB) processes for variable bioenergy quality, (ii) ensuring safe CO₂ transport and storage while mitigating CCS value chain risks, (iii) supporting R&D to reduce costs and enhance efficiency in flexible HB, renewable energy, and storage technologies, and (iv) establishing a technologically neutral policy framework that considers dynamic cost reductions and interactions between policy instruments and technologies.”

2. Literature survey should include recent ammonia production research and energy policy researches. Pls expand.

We recognise that our initial literature review primarily focused on elucidating the fundamental aspects/characteristics associated with incumbent commercial technology options in ammonia production (AP). While this approach provided a solid foundation, it may not have fully captured the latest advancements in novel and promising AP technologies – especially within a US context. Acknowledging the rapid evolution in the field, we have now augmented our literature review to include significant studies in this area that have recently been published.

Two recent studies recently pointed out the same challenges associated with the inflexibility of the conventional Haber-Bosch process (Bose et al., 2022; Bouaboula et al., 2023). In the previous version of our literature review, we did not provide details on the LCOA with different HB flexibilities. The old version reads (Last paragraph of SI section F6.1):

“

Allowing for flexible HB operation involves weighing the costs and benefits of reduced electricity matching requirements against the obvious reduced production, but also the increased risk of damaging process equipment under non optimal operating conditions (Ostuni & Zardi, 2011). Flexibility in HB operation is a prominent cost driver as the CAPEX associated with supplemental renewable energy infrastructure is significantly larger than that of ammonia production or generation of hydrogen. Oftentimes, the minimum load of a flexible HB process is more impactful on overall process system costs than HB CAPEX itself given the impact of minimum load on sizing of supplemental power equipment (Wang et al., 2023). It was determined that a minimum HB load decrease from 60% of nominal to 10% supported by a hybrid renewable energy system (wind and PV) resulted in a 7.1% and 3.9% decrease in LCOA in two Australian locations for which simulations were conducted (Wang et al., 2023).

”

The new version reads as follows:

“

Allowing for flexible HB operation involves weighing the costs and benefits of reduced electricity matching requirements against the obvious reduced production, but also the increased risk of damaging process equipment under non optimal operating conditions (Ostundi & Zardi, 2011). Flexibility in HB operation is a prominent cost driver as the CAPEX associated with supplemental renewable energy infrastructure is significantly larger than that of ammonia production or generation of hydrogen. Oftentimes, the minimum load of a flexible HB process is more impactful on overall process system costs than HB CAPEX itself given the impact of minimum load on the sizing of supplemental power equipment (Wang et al., 2023). It was determined that a minimum HB load decrease from 60% of nominal to 10% supported by a

hybrid renewable energy system (wind and PV) resulted in a 7.1% and 3.9% decrease in LCOA in two Australian locations for which simulations were conducted (Wang et al., 2023). The Bose et al. (2022) study encompasses a cost assessment of a dynamic green ammonia production plant with renewable energy. They take into account the spacio-temporal variation of VRE resources. In particular, Bose et al. (2022) find that 50% capacity flexibility in HB results in a 15% decrease in LCOH. However, this LCOH is not competitive with the incumbent grid-connected technology (\$400/t vs. \$800/t). Bouaboula et al. (2023) performed a techno-economic analysis of a PEM AP system with storage to address steady-state HB constraints. The resulting LCOA in 2020 is \$780/t, in the year 2030 is \$400/t, and in 2050 is \$250/t. Further LCOA reductions may be realised due to solar, wind, and electrolysis-relevant equipment becoming more competitive. As our study points out, Bouaboula et al. (2023) confirm that the cost reductions exhibit non-linear behavior with respect to time.

”

3. *you should explain what is your research implication on energy policy? why do you recommend AP? what is the importance of your research?*

It is true that our paper delves deeply into the economic impacts of the IRA on low-carbon AP, particularly in the Discussion and Conclusions sections. However, we acknowledge the reviewer’s point that explicit policy recommendations based on our findings were not sufficiently addressed.

Indeed, the reviewer’s observation is apt, as our study extends beyond mere economic analysis to encompass policy implications. Recognising this, we have now included specific policy recommendations derived from our research findings in the revised manuscript. These recommendations are designed to guide policymakers in making informed decisions about supporting and advancing demonstration projects for low-carbon AP technologies at the commercial scale. By doing so, we aim to bridge the gap between economic analysis and actionable policy guidance, thereby underscoring the importance and relevance of our research in the broader context of energy policy and sustainable development.

The changes implemented to address this point are distributed across the discussion of the paper. Below is a sample paragraph that was changed.

Before:

“Under the IRA policy framework, both CCS and BH2S prove competitive against SMR, with similar performance levels. The CCS option benefits from a natural market hedge, as ammonia prices closely correlate with natural gas prices. In contrast, BH2S carries a higher market risk since ammonia prices do not correlate with bioenergy costs (refer to SI §G). BH2S’s competitiveness is limited by the costs and availability of relatively inexpensive biomass feedstock. Similarly, the economics of CCS rely on the efficiency of its downstream value chain, including CO₂ transport and storage. Risks or perceptions of CO₂ leakage could adversely affect the IRA’s cash-equivalent support, influencing investment decisions and the adoption rate of CCS technology.

”

After:

“CCS and BH2S are competitive against SMR under the IRA policy framework and perform on par. CCS option has a natural market hedge as ammonia prices are highly correlated with natural gas prices. BH2S has a higher market risk profile than CCS because ammonia price is uncorrelated with bioenergy cost (See SI §G). Further, given its dependence on relatively cheap biomass feedstock, BH2S’s competitiveness is limited by feedstock supply costs and constraints. Similarly, CCS economics depend on the performance of its downstream

value chain – CO₂ transport and storage. Risks (or perception) of CO₂ leakage could negatively impact IRA’s cash-equivalent support, influencing both investment decisions and the rate of CCS technology adoption.

”

We have also expanded more on some intangible risks not quantified in this study that policymakers and investors should consider when leveraging the provisions of the IRA:

“Our analysis assumes a uniform market value for tax credits across all technologies. This assumption is especially relevant for CCS, where environmental and financial risks associated with CO₂ leakage pose substantial challenges (Herzog, 2011; Bartlett & Krupnick, 2019). As per Norton Rose Fulbright, the financial liability for recapture under 45Q credits extends for 15 years (Martin, 2021; Burton, 2023), increasing the risk for tax equity investors. Consequently, the actual market value of these credits might be significantly lower than their face value, affecting investment decisions and technology adoption rates (D’Alelio et al., 2022). Therefore, the presumed effectiveness of tax credits in promoting AP CCS technologies could be more overestimated than other technologies, underscoring the need for a more thorough analysis than what we present here.

AP BH₂S, despite its economic advantages over other LCAP options, has limitations that are not captured in our model. Its reliance on a large, local biomass reserve can be a constraint if the reserve depletes, reducing scalability or may lead to extra costs for in-situ biomass cultivation. This approach raises concerns about significant land use, possibly better suited for agriculture (Arodudu et al., 2020; T.M. Young, 1991). For instance, cultivating algal biomass in ponds costs about \$694-864/MT, whereas our model considers woody biomass costs \$50-118/MT (Klein & Davis, 2021).

”

4. *what is your stochastic economic analysis methodology? and how? Pls explain.*

We acknowledge that the proposed methodological framework in the main manuscript was not comprehensively presented and contained only the elements necessary for the interpretation of the results derived. For replication purposes, we included a more comprehensive presentation of the study’s methodological foundations and framework in the SI (see sections A-E). We also made some revisions to the first paragraph of the Methodology section to adequately address the issue (see below).

Our approach is grounded in a Net Present Value (NPV) analysis through a Discounted Cash Flow (DCF) model to answer the question directly. This model integrates CAPEX and OPEX (O&M) estimates derived from techno-economic analyses sourced from detailed DOE technology reports based on commercial equipment cost quotations. Where data was not directly available, we employed the established methodology outlined by Peters et al. (2003), as explained in the SI. This methodological framework ensures a robust and reliable economic analysis and performance profile assessment in the presence of multiple sources of irreducible uncertainty by capturing both the probabilistic nature of cost variability and the impact of different technological and policy scenarios on the economic feasibility of ammonia production pathways using Monte Carlo simulation techniques. The proposed probabilistic approach overcomes the “flaw of averages” (Savage et al., 2010) traditionally associated with conventional economic performance assessment studies and classical sensitivity analysis (i.e. the fallacy that economic performance evaluated at average baseline conditions represents average economic performance) by generating probabilistically unbiased distribution profiles of costs and NPVs that can also be statistically characterised in an insightful and nuanced manner. Within such a stochastic economic analysis context, identifying and ranking prospects to reduce

exposure to “downside risks” and/or enhance access to “upside opportunities” under various policy support scenarios involving competing technology options becomes feasible.

New version:

“This section first gives an outline of our process economic plant-level model. A comprehensive presentation of the proposed methodological framework is included in the SI (see sections A-E). In particular, we describe the structure and constitutive components of the proposed stochastic DCF model and give an overview of the policy modeling scenarios considered in this study , as well as its key assumptions. All related chemical processes, techno-economic parameters, and assumptions are listed and described in the accompanying SI. Figure 8 depicts the systematic approach applied to every technology option considered in the present study. Table 2 describes the various technologies considered along with their IRA policy eligibility.”

Old version:

“This section first gives an outline of our process economic plant-level model. Then, it describes the stochastic DCF model and gives an overview of policy modeling, scenarios considered in this analysis, and key assumptions. All related chemical processes, techno-economic parameters, and assumptions are listed and described in the accompanying SI. **Error! Reference source not found.** depicts the systematic approach applied to every technology. **Error! Reference source not found.** describes the technologies considered in this study along with their IRA policy eligibility. “

5. *is your research based on theoretical analysis or actual data analysis framework?*

Our study utilises actual data to establish baseline parameters such as equipment costs, raw material consumption, labour expenses, and similar financial metrics. These data points provide an empirically grounded foundational element for our analysis as they are explicitly integrated into the Monte Carlo-based uncertainty quantification methods.

Please note that the structure of the proposed stochastic economic performance assessment framework is also endowed with sound theoretical methodological and analytical foundations in the presence of fundamental uncertainty (please see Remark 4 above).

6. *What are the climate policy implications on the AP processes?*

Our study underscores that under the IRA policy framework, CCS and BH2S emerge as competitive technologies against conventional SMR. This competitiveness is partly attributed to the alignment of these technologies with the objectives of the IRA, which emphasises low-carbon technologies.

However, the AEC process, despite the potential for high public support under the IRA, currently struggles with economic viability. Its success hinges on developing cost-effective clean energy matching rules and substantial technology cost reductions in electrolytic and renewable electricity generation technologies. In 2033, AEC will improve significantly, but not enough to outperform CCS and BH2S.

Furthermore, the interaction between the EU’s CBAM and IRA suggests stronger incentives for decarbonising US ammonia production than with the IRA alone. This could make certain low-carbon AP technologies, like CCS and BH2S, more economically viable, especially under scenarios that involve grid connections.

The study also indicates the need for flexible Haber-Bosch processes to adapt to fluctuating renewable energy inputs, a crucial consideration for future low-carbon AP technologies.

Our research highlights the importance of a technologically neutral policy framework that acknowledges dynamic cost reductions and the interactions between various policy instruments and AP technologies.

These are crucial insights related to energy and climate policies, which are discussed in detail in Section 2, “Results”, with a further discussion and conclusions regarding energy and climate policies in Section 3.

7. What are the constraints of climate policy design which influenced the policy making and economy framework?

Our study modelled these constraints through various policy programs under the Inflation Reduction Act (IRA), such as 45V, 45Q, 45Y, and 48E. These programs offer tax credits based on material production, carbon sequestration, and capital expenditures, significantly influencing the economic viability of different low-carbon AP technologies.

Among these constraints is the dependence of tax credits on each AP technology’s lifecycle carbon intensity (CI). The CI calculation considers multiple components, including direct emissions and upstream emissions from natural gas, biomass, and electricity (see SI C). This approach to quantifying CI determines the eligibility and extent of tax credits under different IRA programs.

Furthermore, the IRA tax credit programs are not directly translatable into actual subsidies due to the complexities of the US tax credit market. This includes factors like transaction costs and the ‘transferability’ of tax credits, which can alter the practical value of these incentives. For instance, the cash-equivalent tax credits received by an AP plant may differ from the nominal value of tax credits due to limitations in income tax abatement and market rates for tax credit exchange (See SI C or Section 2.2 of the main text).

Our study also addresses the interactions between these tax credits and other policy instruments like the EU’s CBAM. This interaction can potentially increase the relative economics of U.S.-based low-carbon ammonia production, especially for technologies like CCS and BH2S.

In summary, the design and modelling of climate policies, particularly tax credits and their interaction with carbon intensities and market mechanisms, form a complex framework that significantly impacts the economic feasibility and decision-making in low-carbon ammonia production.

8. why did you select the HB process for ammonia synthesis?

We appreciate the reviewer’s inquiry regarding our Haber-Bosch (HB) process selection for ammonia synthesis. Our initial manuscript may have inadvertently overlooked a comprehensive discussion of alternative ammonia synthesis methods, such as electrified ammonia production (AP). The HB process remains the predominant method employed globally for commercial-scale ammonia production (IEA, 2021; MacFarlane et al., 2020). While emerging methodologies, including the electroreduction of N₂ and H₂ to NH₃, are thermodynamically promising, they currently encounter limitations in ammonia selectivity, as highlighted by Singh et al. (2019). Given these considerations, the HB process is the most economically viable option.

In response to this valuable feedback, we have amended section 1 of our paper to incorporate this explanation, enhancing the comprehensiveness of our discussion on ammonia synthesis

methods.

New added text:

“The Haber-Bosch (HB) process is preferred over electrochemical methods for ammonia synthesis due to its global prevalence and economic viability for commercial-scale production. While electrochemical methods, like the electroreduction of N₂ and H₂ to NH₃, show thermodynamic promise, they currently face challenges in ammonia selectivity, limiting their practical application compared to the established and efficient HB process (Singh et al., 2019).”

9. *Authors should discuss the economy framework and cost.*

Acknowledging the reviewer’s query regarding the economic framework and cost analysis, it is pertinent to note that the structure and constitutive elements of the proposed methodological framework, detailed in Section 4 and further elucidated in SI Sections A-E, revolves around a robust economic assessment approach. The cornerstone of this assessment is the Net Present Value (NPV) approach, which is applied to evaluate economic performance across various ammonia production technologies. This method integrates a comprehensive monthly cash flow analysis over the ammonia production lifespan, encompassing a spectrum of cost components.

Our approach transcends conventional economic assessments by embedding Monte Carlo simulation techniques within the Discounted Cash Flow (DCF) model, addressing the inherent uncertainties of low-carbon technology options. This stochastic treatment allows for a probabilistically unbiased estimation of expected NPV and other relevant statistical measures, effectively accommodating many uncertainty sources. Complementing this is an in-depth energy and climate policy modelling analysis, dissecting the influence of the Inflation Reduction Act’s incentives on the economic feasibility of different low-carbon ammonia production pathways. The specific mathematics are described in SI sections A-E.

We benchmarked against the IEA’s Ammonia Technology Roadmap to validate the economic model’s robustness. Further, we executed many simulation runs to ensure convergence of the results. The model’s resilience is further underscored by a series of sensitivity analyses probing the impact of various input assumptions on economic outcomes.

10. *The figures caption should be improved.*

The figure captions have been revised to expand on the meaning of the scenarios. See reply to comment 11.

11. *what is the difference between scenario A,B,and C in fig.3? Figures are not clear.*

We agree with the reviewer. We have expanded the figure caption to briefly explain the scenarios and their effect on the carbon abatement cost (CAC). See below.

Old caption:

“NPV versus carbon abatement cost”

New caption:

“NPV versus carbon abatement cost. Scenario A depicts a case with grid emissions – thereby having reduced policy support. Scenario B is an *in-situ* installation of the hybrid wind farm and hence has the largest CAC. Scenario C is where the AP plant has a PPA with the hybrid wind farm – so no tax credits that the wind farm attains are accounted and only AP tax credits are shown.”

12. *what are the implication of energy and climate policy on AP? Pls discuss.*

We thank the reviewer for raising this pertinent question. Below, we include a summary of the topics we touch on when discussing the implications of energy and climate policy on AP. These insights can be found in the discussion section of our paper:

Energy and climate policies significantly impact the AP landscape, mainly through subsidies, carbon pricing, and production rules. The IRA’s support for low-carbon technologies, such as CCS and BH2S, against SMR showcases the vital role of government incentives in altering the competitive dynamics within AP. While CCS enjoys a natural hedge against ammonia price fluctuations linked to natural gas prices, BH2S faces higher market risks due to the mismatch between ammonia prices and bioenergy costs. Such policy instruments are crucial in mitigating investment risks associated with new, low-carbon AP technologies. Furthermore, the valuation of tax credits, particularly for CCS, highlights how environmental and financial risks could influence technology adoption rates and investment decisions by affecting the market value of these credits.

The EU’s ETS and CBAM policies globally underscore a strategic move towards pricing carbon emissions, potentially creating a more conducive environment for low-carbon AP technologies. The interplay between IRA and CBAM could provide more substantial incentives for decarbonising US-based AP, especially for CCS and BH2S technologies, aligning them with the EU’s carbon pricing framework. This synergy between US and EU policies could grant a competitive edge to low-carbon AP technologies, resonating with the EU’s carbon pricing strategy. Additionally, the debate on matching rules for renewable electricity and hydrogen production highlights the policy’s role in defining AP operational dynamics, striving to balance economic feasibility with carbon emission reductions. These policy environments reveal how national and international frameworks critically influence the AP industry’s trajectory towards sustainability.

13. *discussion should be expanded, Pls discuss more.*

As the reviewer suggested, an expansion of the pertinent discussion involving the study’s key findings/conclusions has been incorporated into the revised manuscript. Additionally, a discussion around the model’s assumptions was added.

Reviewer #2 (Remarks to the Author):

This work delivers a stochastic economic analysis of the plant-level ammonia production model considering key low-carbon ammonia production pathways such as steam methane reforming w/ CCS, biomass gasification, and electrolysis. The article comprehensively addresses various aspects, including economic analysis, policy implications, and technology considerations, providing a holistic view of the topic. Overall, the paper deals with a pivotal subject, it is very well written, and the figures are appealing, therefore; this reviewer finds no constraints in advising this article for publishing in Nature Communications journal after some minor revisions.

The scenarios (A, B, C) are outlined effectively, but a brief introductory paragraph summarising their significance could enhance the reader's understanding before delving into the details. Consider using concise bullet points to highlight key features of each scenario.

We thank the reviewer for the valuable remarks and suggestions. In response to the reviewer's suggestion for an introductory paragraph summarising the significance of the scenarios, we have amended the text to enhance clarity and reader comprehension. The revised last paragraph of the introduction now outlines the scenarios (A, B, C).

Old version:

“This study evaluates the economic impacts of the IRA on low-carbon ammonia production (LCAP) technologies at the plant level, focusing on technology, policy, and market uncertainties. We employ a stochastic Discounted Cash Flow (DCF) model to assess the IRA's financial provisions for LCAP: conventional SMR, SMR with Carbon Capture System (CCS), indirect biomass gasification coupled with SMR (BH₂S), and Alkaline Electrolysis (AEC). Unlike traditional DCF models that rely on deterministic cash flow estimates, our approach incorporates uncertainties, addressing the “flaw of averages” (Savage et al., 2010) in the complex, non-linear AP value chain.”

New version:

“This study evaluates the economic impacts of the IRA on low-carbon ammonia production (LCAP) technologies at the plant level, focusing on technology, policy, and market uncertainties. We employ a stochastic Discounted Cash Flow (DCF) model to assess the IRA's financial provisions for LCAP: conventional SMR, SMR with Carbon Capture System (CCS), indirect biomass gasification coupled with SMR (BH₂S), and Alkaline Electrolysis (AEC). Unlike traditional DCF models that rely on deterministic cash flow estimates, our approach incorporates uncertainties, addressing the “flaw of averages” (Savage et al., 2010) in the complex, non-linear AP value chain. The DCF approach will look at three scenarios. Scenario A assumes the grid electricity is used to power the LCAPs and SMR. Scenarios B and C use renewable solar and wind energy and battery storage to power the LCAPs. Scenario B is the case when the AP plant owns the generation facility and scenario C is a power purchase agreement (PPA) business model.”

The explanation of the Monte Carlo simulation approach is commendable. Emphasise the importance of overcoming limitations in traditional models, making it explicit how this methodology enhances the study's robustness.

We modified the text to make the benefit of a Monte Carlo approach more explicit and point to the relevant sections in the SI. These enhancements compare our probabilistic approach to the rest of the literature, where all but one study used deterministic methods.

Old Version:

“These limitations can be effectively overcome by the integration of Monte Carlo simulation techniques that can offer probabilistically unbiased estimates of the expected NPV (or any other performance metric) as well as additional valuable statistical measures (standard deviation, Value at Risk/Value at Opportunity) inferred directly from a detailed characterisation of its probability distribution profile. Furthermore, a Monte Carlo-based assessment framework can simultaneously accommodate multiple sources of uncertainty, unlike traditional sensitivity analysis, where varying a single unknown model input while keeping the rest at nominal/baseline values is the only way to assess the impact on the project’s performance profile.

Our research methodology is designed to quantify variables’ uncertainties in the following categories: market parameters (NG, NH₃, and electricity), policy criteria (48E credits), and each technology’s lifecycle CO₂ intensities (CI). There are also financial inputs, including equity, cost-of-debt, return on equity, loan terms, depreciation, and income taxes – which we keep constant. Market parameters are used to simulate NG, NH₃, and electricity across time using general Brownian motion (calibrated for AEO 2023 for scenario A). ”

“”

New Version:

“These limitations can be effectively overcome by the integration of Monte Carlo simulation techniques that can offer probabilistically unbiased estimates of the expected NPV (or any other performance metric) as well as additional valuable statistical measures (standard deviation, Value at Risk/Value at Opportunity) inferred directly from a detailed characterisation of its probability distribution profile. Furthermore, a Monte Carlo-based assessment framework can simultaneously accommodate multiple sources of uncertainty, unlike traditional sensitivity analysis, where varying a single unknown model input while keeping the rest at nominal/baseline values is the only way to assess the impact on the project’s performance profile. We observe that most techno-economic studies in AP are not probabilistic and have variable results (see SI section F). Our research methodology is designed to quantify uncertainties associated with value-driving model input variables such as market parameters (NG, NH₃, and electricity), policy criteria (48E credits), and each technology’s lifecycle CO₂ intensities (CI). There are also financial inputs, including equity, cost-of-debt, return on equity, loan terms, depreciation, and income taxes – which we keep constant. Market parameters are used to simulate NG, NH₃, and electricity across time using general Brownian motion (calibrated for AEO 2023 for scenario A).”

The sensitivity analyses are well-structured. Please just clearly articulate the rationale behind each sensitivity analysis, emphasising the potential real-world implications and why they are crucial for understanding the economic dynamics of low-carbon ammonia technologies.

We thank the reviewer for the constructive feedback. The rationale behind each sensitivity analysis is clearly articulated in our Methodology section. Acknowledging the reviewer’s point on the real-world implications, we concur that additional clarity is warranted. Consequently, we have implemented changes throughout the discussion section of our paper to enhance the exposition of these real-world implications.

The model validation and robustness section is detailed and provides assurance of the model's credibility. Consider summarising key validation results in the main text and directing the reader to the supplementary files for more in-depth information.

We appreciate the reviewer's positive feedback regarding the model validation and robustness section. In agreement with your suggestion, we have included a footnote at the beginning of the Results section. This footnote directs readers to SI Section G for a detailed exploration of the Monte Carlo results validation, including convergence checks and further validation of our economic analysis through benchmarking against the IEA's Ammonia Technology Roadmap (IEA, 2021). This addition ensures that the critical validation results are briefly summarised in the main text while allowing readers to delve deeper into the specifics within the SI.

The footnote reads:

"We validate our Monte Carlo results by checking for convergence in SI section G. We also validate our economic analysis with a benchmark against the IEA's Ammonia Technology Roadmap (see Section G.2) (IEA, 2021)."

Emphasise the transparency of assumptions, particularly regarding carbon intensity variations and uncertainty sources.

We acknowledge the reviewer's emphasis on transparency. Accordingly, a footnote has been added at the start of the Results section to clarify our assumptions, particularly regarding carbon intensity variations and uncertainty sources. Additionally, the Discussion section now includes a more focused examination of these assumptions.

The new footnote reads:

"We make assumptions and use estimates from the literature to calculate the carbon intensity of the AP technologies. For more information, see section 3.2 and SI section C.3."

[add discussion portion here]

The following article might also be of interest for your references and research: Cardoso, et al., Small-Scale Biomass Gasification for Green Ammonia Production in Portugal: A Techno-Economic Study, Energy&Fuels, 2021, 35, 17, 13847–13862

We thank the reviewer for the reference suggestion. The study by Cardoso et al. has been added to our discussion on AP BH2S.

New text:

"AP BH2S, despite its economic advantages over other LCAP options, has limitations that are not captured in our model. Its reliance on a large, local biomass reserve can be a constraint if the reserve depletes, reducing scalability or may lead to extra costs for in-situ biomass cultivation. This approach raises concerns about significant land use, possibly better suited for agriculture (Arodudu et al., 2020; T.M. Young, 1991). For instance, cultivating algal biomass in ponds costs about \$694-864/MT, whereas our model considers woody biomass costs \$50-118/MT (Klein & Davis, 2021). Furthermore Cardoso et al. (2021) point to biomass availability as

being a key factor in maintaining steady production of ammonia – which they find to be the most important economic factor.

”

Old Text:

“AP BH2S, despite its economic advantages over other LCAP options, has limitations that are not captured in our model. Its reliance on a large, local biomass reserve can be a constraint if the reserve depletes, reducing scalability or may lead to extra costs for in-situ biomass cultivation. This approach raises concerns about significant land use, possibly better suited for agriculture (Arodudu et al., 2020; T.M. Young, 1991). For instance, cultivating algal biomass in ponds costs about \$694-864/MT, whereas our model considers woody biomass costs \$50-118/MT (Klein & Davis, 2021).”

Reviewer #3 (Remarks to the Author):

The authors present a stochastic economic assessment of different avenues for ammonia production.

The work is relevant and results are interesting. However, it would be beneficial to clarify some points. The structure of the work is unconventional, where the premises and assumptions are provided after the results.

A list of acronyms would be very helpful

We thank the reviewer for the helpful suggestion. We have added a table of acronyms at the beginning of the methods section. The same table is pasted below:

Table 1: list of acronyms

Abbreviation	Meaning
CI	Carbon intensity
CAC	Carbon abatement cost
AP	Ammonia production
SMR	Steam-methane reforming
CCS	Carbon capture system
BH2S	Biomass gasification
AEC	Alkaline electrolysis
TC	Tax credit
PTC	Production tax credit
ITC	Investment tax credit
CBAM	Carbon Border Adjustment Mechanism
IRA	Inflation Reduction Act

Regarding Monte Carlo simulations, what tools were used?

Simulations were performed in Python using Numpy's random variable libraries (see methods in the main paper). More details on the implementation of the described methodology and the

program used to generate results will be published on GitHub. They will be available to the public upon publication of the paper.

In NPV calculations, what criteria was followed to set the selling price of product?

In our NPV calculations, the criterion for setting the selling price of the product is based on the market price of ammonia at a given time. Specifically, the selling price is determined using the Geometric Brownian Motion (GBM) model to simulate the market price of ammonia at each time point T. This approach is detailed in SI section C.2.

Why is alkaline electrolysis preferred to other alternatives such as PEM, which present a high degree of flexibility suitable for intermittency of renewable sources?

Alkaline electrolysis, as a longstanding, mature electrolytic hydrogen production technology, offers notable advantages and few idiosyncrasies when operated intermittently. Alkaline electrolyzers are characterised by their cost-effectiveness, mainly due to the use of non-precious metal catalysts. The materials required for their construction are readily available and inexpensive, with common electrode materials using nickel, copper, iron, steel, or titanium and the reactor walls being made of stainless steel and nickel alloys, mitigating potential supply chain disruptions (Brauns & Turek, 2020) and enabling alkaline cells to undercut PEM CAPEX costs by ~55% (Krishnan et al., 2023; Nami et al., 2022) However, alkaline electrolyzers operate at elevated temperatures (80-100°C) with highly caustic electrolytes – requiring frequent maintenance and prescient choice of construction materials resistant to corrosion (Guo et al., 2019).

While these systems exhibit high energy efficiency at stable, high-load durations, alkaline electrolyzers have a low partial-load range due to the numerous unit operations auxiliary to the actual electrolyser that is required to produce a usable hydrogen product (Guo et al., 2019; Krishnan et al., 2023) – i.e. they are less adept at managing the dynamic power inputs characteristic of intermittent renewable energy sources. This limitation is compounded by the relatively lower purity (99.8 wt% vs PEM's 99.99 wt%) of the hydrogen gas they produce, necessitating further purification and drying unit operations to meet the standards required for specific applications (Nami et al., 2022). Additionally, alkaline electrolyzers require stringent control over process conditions for proper operation, which can be challenging in the face of fluctuating power input. In short, alkaline electrolysis exemplifies a low capital cost, higher maintenance, and higher footprint of hydrogen electrolysis technology, which operates best under steady-state conditions. However, its lower upfront cost compared to PEM may allow it to succeed in settings where demand elasticity and capacity fluctuations are relatively limited and space for battery storage is not at a premium.

In contrast, Proton Exchange Membrane (PEM) electrolyzers excel in their ability to rapidly adjust to the variable power outputs characteristic of renewable energy sources, along with their capacity to produce high-purity hydrogen within a significantly smaller footprint relative to alkaline electrolyzers (which is particularly relevant in systems requiring more real estate for peak shaving storage systems) (Guo et al., 2019; Nami et al., 2022). The hydrogen produced via PEM electrolyzers is of high purity (99.99%) (Guo et al., 2019). It only uses deionised water,

no longer necessitating intense corrosion resistance and further reducing space requirements for the purification and alkaline regeneration unit operations required of alkaline electrolyzers. Their lower operating temperature also increases their responsiveness to sudden shifts in power capacity.

PEM's operational flexibility complements the variable output of renewable energy sources. However, their reliance on costly materials such as expensive sulfonated polymer separators and precious metal catalysts inflates both the capital cost and the complexity of maintenance; these catalysts are prone to corrosion and the membrane degrades – and both mechanisms are exacerbated under dynamic conditions (Papakonstantinou et al., 2020). This necessitates frequent replacements, ultimately limiting the system's lifespan to nearly half (3-5 years) of an alkaline electrolyser (~10 years) (Guo et al., 2019). While the operational flexibility of PEM electrolyzers is advantageous for managing variable loads and ensuring compatibility with renewable energy sources, the associated higher costs and concerns about long-term durability under intermittent operation emphasise the need to understand their operation in the context of a more extensive integrated system to characterise their economic viability and operational robustness better. To this end, the difference between the lower capital cost of alkaline and intermittency-robust PEM electrolysis technologies may attenuate when considering auxiliary systems such as using peak-shaving storage for alkaline electrolysis to manage its sensitivity to partial loads.

In our analysis, hydrogen is steadily produced for every technology option. Integrating a battery storage system guarantees a stable and adequate electricity supply, which is crucial for synthesising the steady supply of syngas to the Haber-Bosch process. When comparing PEM and AEC within this context, AEC emerges as more advantageous due to the absence of a requirement for variable production. AEC's lower capital cost, ranging from \$500-1200/kW compared to PEM's \$1800-800/kW reinforces its favorability. AEC also has a longer lifetime than PEM (80,000hrs vs. 60,000hrs) (Schmidt et al., 2017).

In a scenario where AEC operates continuously, and PEM's output is variable, the suitability of each technology becomes less clear-cut. PEM could be more advantageous if hydrogen storage costs are minimal. Additional costs from hydrogen storage costs originate from the requirement for specialised materials and technologies for safe storage under high pressures or at cryogenic temperatures (MacFarlane et al., 2020). Additionally, the energy demands for hydrogen compression and the establishment of an extensive storage and refuelling infrastructure significantly contribute to these expenses, which is a critical value risk given the steady hydrogen supply needed for the Haber-Bosch process.

PEM's flexibility offers a competitive advantage by enabling hydrogen storage options. However, for PEM to be more competitive than AEC, the cost of hydrogen storage must not only be at least lower than that of battery storage but also need to offset the cost difference between PEM and AEC. This cost balance is crucial for PEM to emerge as the more economically viable option. For a back-of-the-envelope estimate of viability, we compare the AEC versus the PEM system CAPEX. First, we check if the cost of hydrogen storage is less than the cost of battery storage:

Battery storage costs approximately 206 \$/kWh, and hydrogen storage costs 11.49 \$/kWh (Abdin et al., 2022; Cole & Frazier, 2019). For simplicity, we converted electrical kWh requirements from hydrogen-equivalent ones using the hydrogen LHV (33 kWh/kg) and

assumed an efficiency of 70% for both processes. We take the average cost of AEC and PEM to be 600 \$/kW_{h2} and \$975/kW_{h2}. We posit these cost estimates:

$$C_{PEM} = 975 + r * 11.49$$

$$C_{AEC} = 600 + r * 206$$

where the parameter r is the ratio of storage to production capacities. PEM would only choose hydrogen storage in an optimal system since it is much cheaper than battery storage. Since we assume steady-state operation, hydrogen storage is not reasonable for AEC. The intersection point for these two lines is at a ratio of 1.9 – where higher ratios favour PEM, and lower ratios favour AEC.

If the operational costs of PEM and AEC are the same, then a storage-to-production capacity ratio of 1.9 would result in equal performance for both technologies. Regarding production, AEC is known to have a higher operational expense, while PEM is cheaper. In terms of storage, hydrogen storage has considerable operational expenses in comparison to battery storage (Abdin et al., 2022). Palys & Daoutidis (2020) studied a PEM and battery storage system and found the optimal LCOA to be \$774/t. This design only includes battery storage to power the HB process and lets the electrolyser and ASU vary flexibly with VRE resources. This study did not compare their performance to AEC —which is competitive with other green energy alternatives (Bose et al., 2022; Bouaboula et al., 2023).

The advantages and disadvantages of each technology are clear. In a future study, we intend to examine the comparative value of PEM versus AEC, explicitly focusing on assessing their flexibility in various hydrogen end-use industries such as steel, ammonia, methanol, and oil refining under the IRA. This analysis will provide crucial insights into the optimal applications of these technologies across different sectors.

A point to discuss is the scale which can be aimed with different primary energy sources. This can also present a relevant competitive advantage for natural gas based SMR in terms of economies of scale.

We thank the reviewer for highlighting the importance of considering different primary energy sources' scale and economic dynamics, particularly concerning natural gas-based technologies. In response, we have expanded our analysis to include a scenario reflecting the higher natural gas prices typical of the European market centred around \$15/mmBTU (See section G.4 of the SI).

Indeed, the scale that AP SMR and AP CCS can achieve is greater than that of AP BH2S because of the supply constraints of biomass and the existing and robust natural gas infrastructure. With AP AEC, the availability of a nearby water reserve may significantly affect scalability – either with prohibitive osmosis costs or water transportation costs. We discuss some scale limitations within the discussion section of the paper. We highlight the importance of AP SMR as a difficult benchmark in the introduction through this new paragraph:

“AP SMR capitalises on the existing robust natural gas infrastructure and highly mature technology, providing it with a significant economic edge over emerging and risky technologies (IEA, 2021). The inclusion of AP SMR as a benchmark in our analysis, an important detail not covered in every study, establishes a high benchmark for alternative technologies aiming to enter the ammonia commodity market.”

Regarding the discussion about matching, given that wind presents significant seasonal fluctuations, is it not an argument to consider a yearly span in the evaluation. Please clarify the methodology.

We concur with the reviewer's comment regarding our study's energy-matching approach. In our analysis, we adopt a yearly matching scenario where the total annual energy output from the wind farm aligns with the yearly energy requirement for operating the ammonia production (AP) plants. This method effectively neutralises the impact of seasonal fluctuations on the energy supply. For a detailed exploration of this approach and its implications, we invite readers to refer to the sensitivity analysis in Section 3, Supplementary Information (SI) Sections D and E, and the discussions section of our study.

Some insights regarding natural gas costs that would render the renewable route more attractive would be interesting.

We consider a sensitivity analysis of natural gas prices on the performance of the plants. However, this only considers US natural gas prices, which are low by global standards. We expanded the sensitivity analysis range to reach EU/Asia-level natural gas import costs faced in 2022 (see SI X).

Why is the renewable resource only limited to wind? Solar PV presents very attractive costs that can improve competitiveness of AEC.

In response to the insightful comments provided by the reviewer, we have undertaken a significant revision of our analysis to encompass not just wind energy resources but also the promising potential of solar photovoltaic (PV) systems. This revision was propelled by the recognition that our initial analysis might have presented a constrained perspective on renewable energy opportunities for AP AEC. By expanding our scope to include solar PV, we aim to offer a more holistic view of renewable energy's potential to enhance AEC's competitiveness and sustainability.

Our revised analysis now thoroughly evaluates the suitability of various locations for solar energy deployment alongside wind energy, as initially reported by Nutrien. This expanded evaluation involved an exhaustive check of all locations highlighted in the Nutrien report ((Nutrien, 2023), ensuring a comprehensive inclusion of viable sites for renewable energy projects. Furthermore, we optimised the electricity generation plants at each location using capacity factor (CF) data from the 2019 Ninja Renewables dataset, enhancing our projections' accuracy and recommendations.

Significantly, our analysis now assumes an advanced make of wind turbines by 2030, drawing on inputs from the DOE reports on the design specifications of wind turbines projected for 2023 and 2030 (see SI D). However, for solar PV, we have maintained the 2023 and 2030 specifications unchanged due to the unavailability of technology specifications for wind. This approach would set a conservative projection for solar PV.

In addressing the methodological framework for our analysis, we acknowledged the necessity to refine our stochastic method for accounting for location-specific renewable energy potentials. Initially, our model uniformly distributed solar, wind, and battery storage capacities between the minimum and maximum optimal capacities across locations. This approach led to independent distributions of capacities, sometimes resulting in simulations that did not adhere to the constraints of our optimisation process.

To rectify this, we adopted a more precise and location-sensitive method. Our optimisation now occurs distinctly for each location, allowing the model to randomly select a location and utilise the optimal capacity specific to that locale. This refined method ensures that the capacities introduced into the AP model strictly satisfy the optimisation constraints, thereby enhancing the reliability and accuracy of our projections.

Through these revisions and methodological enhancements, our analysis presents a more accurate and comprehensive assessment of the potential for renewable energy to bolster the competitiveness of AP AEC. We believe that including solar PV, alongside refined analytical techniques, significantly enriches our understanding of renewable energy's role in advancing sustainable and competitive energy solutions.

References

- Abdin, Z., Khalilpour, K., & Catchpole, K. (2022). Projecting the levelized cost of large scale hydrogen storage for stationary applications. *Energy Conversion and Management*, 270, 116241. <https://doi.org/10.1016/j.enconman.2022.116241>
- Bose, A., Lazouski, N., Gala, M. L., Manthiram, K., & Mallapragada, D. S. (2022). Spatial Variation in Cost of Electricity-Driven Continuous Ammonia Production in the United States. *ACS Sustainable Chemistry & Engineering*, 10(24), 7862–7872. <https://doi.org/10.1021/acssuschemeng.1c08032>
- Bouaboula, H., Ouikhalfan, M., Saadoun, I., Chaouki, J., Zaabout, A., & Belmabkhout, Y. (2023). Addressing sustainable energy intermittence for green ammonia production. *Energy Reports*, 9, 4507–4517. <https://doi.org/10.1016/j.egy.2023.03.093>
- Brauns, J., & Turek, T. (2020). Alkaline water electrolysis powered by renewable energy: A review. In *Processes* (Vol. 8, Issue 2). <https://doi.org/10.3390/pr8020248>
- Cole, W. J., & Frazier, A. (2019). *Cost Projections for Utility-Scale Battery Storage*. <https://doi.org/10.2172/1529218>
- Guo, Y., Li, G., Zhou, J., & Liu, Y. (2019). Comparison between hydrogen production by alkaline water electrolysis and hydrogen production by PEM electrolysis. *IOP Conference Series: Earth and Environmental Science*, 371(4), 042022. <https://doi.org/10.1088/1755-1315/371/4/042022>
- IEA. (2021). Ammonia Technology Roadmap; Towards more Sustainable Nitrogen Fertiliser Production. *International Energy Agency*.
- Krishnan, S., Koning, V., Groot, M., Groot, A., Mendoza, P. G., Junginger, M., & Kramer, G. J. (2023). Present and future cost of alkaline and PEM electrolyser stacks. *International Journal of Hydrogen Energy*, 48(83), 32313–32330. <https://doi.org/10.1016/j.ijhydene.2023.05.031>
- MacFarlane, D. R., Cherepanov, P. V, Choi, J., Suryanto, B. H. R., Hodgetts, R. Y., Bakker, J. M., Ferrero Vallana, F. M., & Simonov, A. N. (2020). A Roadmap to the Ammonia Economy. *Joule*, 4(6), 1186–1205. <https://doi.org/https://doi.org/10.1016/j.joule.2020.04.004>
- Nami, H., Rizvandi, O. B., Chatzichristodoulou, C., Hendriksen, P. V., & Frandsen, H. L. (2022). Techno-economic analysis of current and emerging electrolysis technologies for green

hydrogen production. *Energy Conversion and Management*, 269, 116162.
<https://doi.org/10.1016/j.enconman.2022.116162>

Nutrien. (2023). *2023 Fact Book*. Nutrien.

Ostundi, R., & Zardi, F. (2011). *Method for load regulation of an ammonia plant* (Patent US9463983B2).

Palys, M. J., & Daoutidis, P. (2020). Using hydrogen and ammonia for renewable energy storage: A geographically comprehensive techno-economic study. *Computers & Chemical Engineering*, 136, 106785. <https://doi.org/10.1016/j.compchemeng.2020.106785>

Papakonstantinou, G., Algara-Siller, G., Teschner, D., Vidaković-Koch, T., Schlögl, R., & Sundmacher, K. (2020). Degradation study of a proton exchange membrane water electrolyser under dynamic operation conditions. *Applied Energy*, 280, 115911. <https://doi.org/10.1016/j.apenergy.2020.115911>

Schmidt, O., Gambhir, A., Staffell, I., Hawkes, A., Nelson, J., & Few, S. (2017). Future cost and performance of water electrolysis: An expert elicitation study. *International Journal of Hydrogen Energy*. <https://doi.org/10.1016/j.ijhydene.2017.10.045>

Wang, C., Walsh, S. D. C., Longden, T., Palmer, G., Lutalo, I., & Dargaville, R. (2023). Optimising renewable generation configurations of off-grid green ammonia production systems considering Haber-Bosch flexibility. *Energy Conversion and Management*, 280, 116790. <https://doi.org/10.1016/j.enconman.2023.116790>

REVIEWERS' COMMENTS

Reviewer #1 (Remarks to the Author):

Congrats to the authors, they covered all comments and the editors can proceed to the publication process.

Reviewer #2 (Remarks to the Author):

All the required modifications have been added. This reviewer has nothing more to add to this work. This work is now up to standard.

Reviewer #3 (Remarks to the Author):

The authors have effectively addressed the questions posed and I can recommend publication of their work.